# Quantification of post-glacier bedrock surface erosion in the European Alps using [10]Be and OSL exposure dating.

Joanne Elkadi[1], Benjamin Lehmann[2], Georgina King[1], Olivia Steinemann[3], Susan Ivy-Ochs[3], Marcus Christl[3] and Frederic Herman[1]

[1]Institute of Earth Surface Dynamics, University of Lausanne, 1015 Lausanne, Switzerland
[2]INSTAAR and Department of Geological Sciences, University of Colorado Boulder, Boulder, CO 80309, USA
[3]Laboratory of Ion Beam Physics, ETH Zürich, Otto-Stern-Weg 5, 8093 Zürich, Switzerland

*Correspondence to:* Joanne Elkadi (joanne.elkadi@unil.ch)

**Abstract.** The retreat of glaciers since the Last Glacial Maximum in the European Alps has left an imprint on topography through various erosional processes. However, few methods are currently capable of resolving these mechanisms on Lateglacial to Holocene timescales. Quantifying the relative contributions of mountain erosion, during these different climate cycles, is useful for understanding long-term landscape evolution and the links between global climate and erosion. Here, we combine three Optically Stimulated Luminescence (OSL) exposure dating signals with [10]Be surface exposure dating to constrain the post-glacier erosion rates of bedrock samples adjacent to the Gorner glacier in the European Alps. The results reveal erosion rates on the order of $10^{-2}$ to $10^{-1}$ mm a$^{-1}$, in general agreement with other studies in the region, as well as a strong negative correlation between erosion rate and elevation implying that frost crack weathering is perhaps not the dominant form of post-glacier weathering. Finally, a global compilation of both subglacial and periglacial erosion rates shows that periglacial erosion rates could be greater than previously thought. Yet, subglacial erosion remains higher, implying that it continues to have the stronger influence on shaping landscapes. Therefore, with a changing climate periglacial erosion rates are likely to remain transient. These insights could lead to important implications for landscape evolution models.

## 1. Introduction

The interplay between erosion and climate has sparked debates and inspired research aimed at better understanding the efficacy of various erosion mechanisms on long-term landscape evolution, as well as the role that climate, and its variability, plays in setting these erosion rates (e.g. Zhang et al., 2001; Molnar, 2004; Willenbring and von Blanckenburg, 2010; Lupker et al., 2013; Cogez et al., 2015; Herman et al., 2013; Herman and Champagnac, 2016; Willenbring and Jerolmack; 2016). Globally, continental topography has been shaped partly through erosional processes associated with rivers, glaciers, soils, rock fall and weathering. For high mountain environments specifically, the strong imprint of glacial and non-glacial erosion is observed at mid- to high-latitudes, but their specific mechanisms and respective impacts on the topography remains convoluted (e.g. André, 2002a; Ballantyne, 2002; Koppes and Montgomery, 2009). Here, non-glacial erosion refers broadly to any erosion occurring in a glacial environment that is not related to subglacial erosion. It is necessary to better quantify this to develop our knowledge of the influence of mountain erosion on the global feedback loop that exists between climate and erosion during glacial and interglacial times.

Currently in alpine environments, glacial erosion and its associated processes is thought to play a dominant role, and thus extensive research has addressed its quantification, as well as the timing of deglaciations (e.g. Hallet et

al., 1996; Montgomery, 2002; Ivy-Ochs and Briner, 2014; Herman et al., 2015; 2018; Wirsig et al., 2016a; b; 2017; Ruszkiczay-Rüdiger et al., 2021; Steinemann et al., 2021). In contrast, studies exploring periglacial erosion or erosion during interglacial times have mainly investigated local fluvial incision (e.g. Korup and Schlunegger, 2007; Valla et al., 2010; Rolland et al., 2017) or catchment-wide erosion rates. A small number of studies have successfully quantified periglacial and interglacial erosion rates from bedrock surfaces through novel techniques (e.g. Kirkbride and Bell, 2010; Sohbati et al., 2018; Smedley et al., 2021), which is discussed in further detail in Section 1.1. Nevertheless, disentangling the relative contributions of the various erosional processes remains challenging (e.g. Hallet et al., 1996; Delmas et al., 2009; O'Farrell et al., 2009; Guillon et al., 2015; Cook et al., 2020).

We extend this dataset by applying the recently developed approach from Lehmann et al. (2019), that combines Beryllium-10 ($^{10}$Be) Terrestrial Cosmogenic Nuclide (TCN) dating with Optically Stimulated Luminescence (OSL) surface exposure dating, to investigate bedrock post-glacier erosion rates (i.e. erosion since glacier retreat). This was done for six samples down a vertical transect adjacent to the Gorner glacier near Zermatt, Switzerland. Then, we examine any potential trends between elevation or slope with our erosion rate results and find a strong negative correlation between erosion rate and elevation, but no correlation between erosion rate and surface slope. Finally, the post-glacier erosion rates from this study are combined with global studies of both subglacial and periglacial erosion rates to reveal that periglacial erosion rates could be more comparable to subglacial erosion rates than anticipated.

## 1.1  Measuring erosion rates in deglaciated environments

At present, there exists a wide range of analytical techniques capable of quantifying bedrock erosion rates across different time intervals (please refer to Moses et al., 2014 and Turowski and Cook, 2017 for in-depth reviews). For timescales on the order of seconds to decades, these methods can include remote sensing (e.g. photogrammetry on both small and large spatial scales; Inkpen et al., 2000; Dornbusch et al., 2008), or measurements relative to anthropogenic reference points (e.g. lettering on gravestones; Inkpen and Jackson, 2000). On the other hand, studies targeting longer timescales ($> 10^3$ years) have measured relative to natural reference points (e.g. resistant quartz veins; Dahl, 1967; André, 2002b; Nicholson, 2008), exploited the half-lives of different cosmogenic nuclides (e.g. Nishiizumi et al., 1986; Bierman and Caffee, 2002; Balco et al., 2008) or used thermochronometry (e.g. Reiners and Brandon, 2006; Herman and King, 2018). Unfortunately, there is a lack of available methodological approaches to quantify bedrock erosion rates across the intermediate time interval, which has recently driven focussed research to devise new methods capable of doing so (Sohbati et al., 2018; Brown and Moon, 2019; Lehmann et al., 2019).

A small number of studies worldwide have already attempted to calculate periglacial rock surface erosion rates and have yielded a wide range of results. These include an investigation using TCN in the western US mountain ranges that estimated the maximum surface erosion rates of alpine bedrock summits at $7.6 \times 10^{-3}$ mm a$^{-1}$ (Small et al., 1997), in contrast to another TCN study in the Nepal high Himalayas which instead found erosion rates of $8 \times 10^{-2} - 2 \times 10^{-1}$ mm a$^{-1}$ (Heismath and McGlynn, 2007). In Europe, using reference quartz veins in Norway found erosion rates of $5.5 \times 10^{-4}$ mm a$^{-1}$ (Nicholson, 2008), while incorporating the edge roundness of boulders in Scotland produced erosion rates of $3.3 \times 10^{-3}$ mm a$^{-1}$ (Kirkbride and Bell, 2010). Recent studies combining TCN

and OSL surface exposure dating in the Eastern Pamirs, China, and the Mont Blanc Massif, France, revealed bedrock surface erosion rates of <3.8 x $10^{-5}$ and 1.72 x $10^{-3}$ mm $a^{-1}$ (Sohbati et al., 2018) and 3.53 x $10^{-3}$ – 4.3 mm $a^{-1}$ (Lehmann et al., 2019, 2020), respectively. Smedley et al. (2021) also combined TCN and OSL surface exposure dating in NW Scotland to derive interglacial erosion rates over the last 4 ka that were consistent with local independent erosion rate estimates (Kirkbride and Bell, 2010), and further inferred that some of their results could be explained by climatic fluctuations that are known to have occurred over that time period. Finally, a global compilation calculated by Portenga and Bierman (2011) gave an erosion rate of 1.2 x $10^{-2}$ mm $a^{-1}$ by averaging the results from studies that applied $^{10}$Be to bedrock surfaces. Here, we use a newly developed approach (Lehmann et al., 2019) that combines two surface exposure dating methods- $^{10}$Be and OSL- to investigate bedrock post-glacier erosion rates and onset times. In this case, the definition of erosion will be the removal of bedrock surface material. Bedrock surfaces offer great potential for the quantification of post-glacial erosion as: (1) they are almost instantaneously exposed to the atmosphere once the ice retreats, and thus immediately begin to record any changes to the surface associated with post-glacial erosion and (2) they are relatively durable in nature rendering them capable of recording long-term erosional histories.

TCN are formed at or near the Earth's surface within specific target minerals as a result of the Earth's constant bombardment by high energy cosmic rays (Dunai, 2010; Gosse and Phillips, 2001). Consequently, following exposure, the concentration of nuclides measured in bedrock can be converted into an apparent exposure age. In this study, we focus on measuring $^{10}$Be which is found in quartz. In contrast, OSL is a trapped charge dating technique where a mineral, such as quartz or feldspar, emits light upon light stimulation due to electrons trapped in defects in the mineral's crystal lattice (Huntley et al., 1989; Aitken, 1998). The intensity of the light emitted is an indication of the concentration of trapped electrons. In recent years, the application of OSL to rock surface exposure dating has proved successful in a variety of settings (e.g. Sohbati et al., 2015; Lehmann et al., 2018; Liu et al., 2019) and is based on the principle that, for an exposed surface, the sun's energy is sufficient to naturally reduce the surface luminescence signal to zero (e.g. Sohbati et al., 2011, 2012). This phenomenon is termed "bleaching". Due to the attenuation of light, this bleaching effect decreases exponentially with depth into the rock until it becomes negligible (Habermann et al., 2000; Polikreti et al., 2002, 2003; Laskaris and Liritzis, 2011). Nonetheless, studies have shown that this depth of bleaching increases with exposure time (Habermann et al., 2000; Polikreti et al., 2002, 2003; Laskaris and Liritzis, 2011; Sohbati et al., 2011, 2012; Lehmann et al., 2018; Gliganic et al., 2019; Sellwood et al., 2019) and, after calibration to account for rock-specific bleaching rates, this bleaching depth can be translated directly into an apparent exposure age for surfaces which have not been affected by erosion (e.g. Lehmann et al., 2018; Sohbati et al., 2018). Recent luminescence instrument developments (Lapp et al., 2015) have facilitated the measurement of rock slices without requiring further mineral separation. Multiple luminescence signals can now be measured from the same slice to obtain the maximum amount of information (e.g. Jenkins et al., 2018; Luo et al., 2018; Meyer et al., 2018; Elkadi et al., 2021; Smedley et al., 2021). Although both TCN and OSL surface exposure dating are influenced by exposure time, they are also affected by surface erosion, and if this is not accounted for then it can lead to an underestimation of exposure ages (e.g. Gosse and Phillips, 2001; Lehmann et al., 2019, 2020). However, the two methods have different sensitivities to erosion (Sohbati et al., 2018; Lehmann et al., 2019) since TCN are formed several metres below a rock surface (Lal, 1991) whereas bleaching fronts in OSL depth profiles are created only in the top mm to cm (Vafiadou et al., 2007;

Sohbati et al., 2011, 2012; Freiseleben et al., 2015). By using these two techniques in conjunction, this difference in sensitivity can be exploited to calculate the surface erosion of bedrock.

## 1.2 Study area

The post-glacier erosion rates were calculated from the flanks of the Gorner glacier, located near the village of Zermatt, Switzerland. This area was chosen due to its well constrained glacial history, consisting of a rich collection of geological maps and aerial photos as well as human observations, but also as a result of its proximity to the only other study that applied this method in the Western Alps (Lehmann et al., 2019, 2020) allowing for direct comparisons.

Bedrock material was collected from six sampling sites down a vertical transect, with sample lithologies consisting of hornfels, schist and gneiss (Table 1). Geomorphological reconstructions (Bini et al., 2009) suggest that, aside from the highest elevation sample, the transect was covered in ice during the Last Glacial Maximum (LGM) and has been de-glaciated since. The three lower elevation samples (GG04, GG05 and GG06) had additional exposure age information from old maps and photos acquired from the Swiss Federal Office of Topography. The three uppermost samples exhibited significant weathering, whereas the three lower elevation samples had extremely well-preserved glacial morphologies and striations (Figs. 1, S1). Samples were collected from bedrock using a combination of a hammer, chisel and Husqvarna K760 power cutter with a diamond blade. Between two to four blocks with dimensions of ≈15 cm × 15 cm x 10 cm were extracted at each site, allowing for a sufficient amount of material for both OSL and [10]Be surface exposure dating.

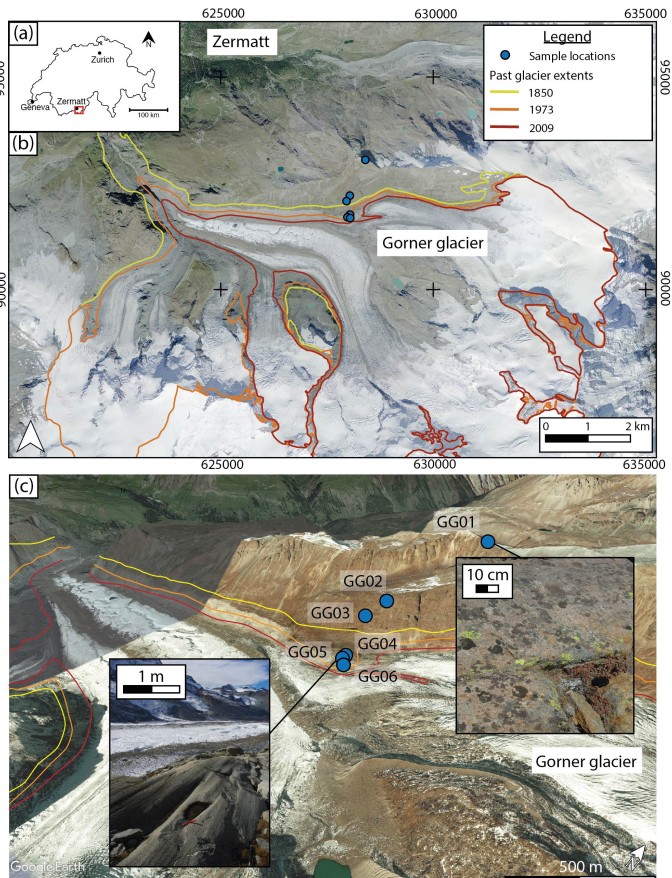

**Figure 1: Study area and sampling sites on (a) regional, (b) glacier basin and (c) local scales. Figure 1(b) further depicts the glacier retreat history obtained from the Global Land Ice Measurements from Space initiative (GLIMS) (GLIMS Consortium, 2005; Raup et al., 2007). The photo in the background is from the Swiss Federal Office of Topography and was taken in 2015. 1(c) illustrates the difference in surface preservation with elevation, and thus exposure age.**

## 2. Methodology

### 2.1 [10]Be sample preparation, measurement and age calculation

Sample preparation for [10]Be dating was undertaken at ETH Zurich, Switzerland, and began with isolating quartz from the bulk rock. To do this, the uppermost 4 cm of each sample was crushed and sieved to obtain grain sizes between 250 and 1000 μm, before being passed through a Frantz magnetic separator and subsequently treated with HCl and a low concentration HF solution. Once pure quartz was isolated and dried, extraction of [10]Be followed the well-established procedure outlined in Kohl and Nishiizumi (1992) and Ivy-Ochs (1996). First, the quartz grains were spiked with 0.25 mg of a [9]Be carrier and dissolved with supra pure HF (40%). Samples were then purified using two ion exchange resins to remove unwanted anions and cations followed by the final selective pH precipitation of $Be(OH)_2$. [10]Be/[9]Be ratios of the samples were measured using the 500 kV TANDY system at the accelerator mass spectrometry facility of ETH Zürich, Switzerland, using the in-house standard S2007N (Christl et al., 2013) calibrated against the 07KNSTD standard (Nishiizumi et al., 2007).

Surface exposure ages were determined using a modified version of the CREp online calculator (Martin et al., 2017; Lehmann et al., 2019) allowing for the application of a step-wise erosion rate correction. The calculations were done with blank corrected [10]Be/[9]Be ratios (full chemistry long-term laboratory procedural blank of [10]Be/[9]Be $(3.2 \pm 1.7)$ x $10^{-15}$), the rescaled SLHL Chironico landslide local production rate ($4.16 \pm 0.10$ atoms $g_{qtz}^{-1}$ $a^{-1}$; Claude et al., 2014) corrected for each sample's latitude, longitude and elevation, the Lifton-Sato-Dunai (LSD) scaling scheme (Lifton et al., 2014), the ERA40 atmospheric model (Uppala et al., 2005) and the Lifton VDM 2016 geomagnetic database (Pavon-Carrasco et al., 2014 for ages between 0 and 14 ka; GLOPIS-75, Laj et al., 2004 for ages between 14 and 74 ka). Further information regarding the input data can be found in Table 1.

## 2.2 OSL

### 2.2.1 Sample preparation and measurement

Luminescence sample preparation followed the method described in Lehmann et al. (2018) and was carried out at the University of Lausanne, Switzerland. Following collection, samples were immediately placed into black, light obstructing bags and all ensuing laboratory work was done under subdued, red-light conditions. Laboratory work began with coring the samples using a water-cooled Husqvarna DM220 drill to extract multiple cores per sample, each with a diameter of 10 mm. Attention was made to drill as far from the edges as possible, to avoid any potential signal resetting that may have occurred during fieldwork, and from areas with minimal lichen cover and red, iron-oxide staining that would have otherwise impeded light penetration and impacted the luminescence signal. The cores were then cut into thin slices ($\approx$0.7 mm thickness) using a Buehler IsoMet low-speed saw mounted with a 0.3 mm thick diamond blade and in the presence of a lubricant. The exact thickness of each slice was measured using a TESA Digitcal Caliper to allow precise reconstructions of the luminescence signal with depth. Example disc slices can be found in Figure S2.

Luminescence measurements followed the protocol outlined in Table 2, with low heating rates, extended isothermal holding times of samples prior to stimulation (1°C s$^{-1}$ and 100 s respectively; Jenkins et al., 2018), and fragments of the slices placed in metal cups during measurement (Elkadi et al., 2021). Three signals were measured per sample- the $IRSL_{50}$ and post-IR $IRSL_{225}$ signals from feldspar and the post-IR $IRSL_{225}$ $OSL_{125}$ (hereinafter referred to as $OSL_{125}$) signal predominantly from quartz. This was done to extract the maximum amount of information possible from each sample as the signals vary in bleaching rates. Since quartz and feldspar

minerals are best stimulated using different wavelengths, different filters were required to isolate the luminescence emissions from the simulation wavelengths (Table 2). All measurements were performed using Risø TL-DA 20 TL-OSL readers (Bøtter-Jensen et al., 2010) equipped with a DASH head (Lapp et al., 2015) and [90]Sr/[90]Y beta source. The environmental dose rate ($\dot{D}$) was calculated from U, Th, K and Rb concentrations of the bulk rock

sample determined using ICPMS at Actlabs, Canada, and the DRAC online calculator (Durcan et al., 2015). All the luminescence signals were subsequently screened using three acceptance criteria: (1) maximum error of the test dose signal ($T_n$) < 15%, (2) $T_n$ greater than 3σ above the background signal and (3) monotonic signal decay indicative of good heating (Elkadi et al., 2021). Any slices which did not meet these criteria were excluded from further analysis as their results were considered not reproducible.

**Table 2: Protocol used for measuring the luminescence signals in the rock slices.**

| Stimulation | Filter | Emission wavelength* | Signal | Target mineral |
|---|---|---|---|---|
| Preheat at 250 °C for 100 s | BG39 + BG3 | | | |
| IRSL at 50 °C for 200 s | BG39 + BG3 | Violet (410 nm) | IRSL$_{50}$ Ln | Feldspar |
| IRSL at 225 °C for 200 s | BG39 + BG3 | Violet (410 nm) | post-IR IRSL$_{225}$ Ln | Feldspar |
| OSL at 125 °C for 200 s | U340 7.5 mm | Near UV (360 nm) | OSL$_{125}$ Ln | Quartz |
| Test dose 51.75 Gy | | | | |
| Preheat at 250 °C for 100 s | BG39 + BG3 | | | |
| IRSL at 50 °C for 200 s | BG39 + BG3 | Violet (410 nm) | IRSL$_{50}$ Tn | Feldspar |
| IRSL at 225 °C for 200 s | BG39 + BG3 | Violet (410 nm) | post-IR IRSL$_{225}$ Tn | Feldspar |
| OSL at 125 °C for 200 s | U340 7.5 mm | Near UV (360 nm) | OSL$_{125}$ Tn | Quartz |

* value represents wavelength that the emission is centred on.

### 2.2.2 Constraining the surface exposure dating model

The evolution of a luminescence signal $L(x,t,r')$ (dimensionless) into a rock surface for a given depth $x$ (mm), time $t$ (year) and recombination centre distance $r'$ (dimensionless) can be modelled using the differential equation below (Lehmann et al., 2019):

$$\frac{dL(x,t,r')}{dt} = \frac{\dot{D}}{D_0}[1 - L(x,t,r')] - L(x,t,r')\overline{\sigma\varphi_0}e^{-\mu x} - L(x,t,r')se^{-\rho'^{-\frac{1}{3}}r'} + \dot{\varepsilon}(t)\frac{dL(x,t,r')}{dx} \qquad (1)$$

This equation describes the four competing processes occurring following a surface's exposure to daylight: (1) electron trapping as a result of ambient radiation, (2) optical electron detrapping due to daylight exposure (bleaching), (3) athermal electron detrapping of the IRSL signal, most likely from quantum mechanical tunnelling

in feldspars (Huntley, 2006; Kars et al., 2008) and (4) surface erosion. Definitions of the symbols can be found in Table 3, and we refer to Lehmann et al. (2019) for further descriptions. A $D_0$ value of 500 Gy (Lehmann et al., 2019) was selected for all samples following sensitivity tests that revealed the negligible effect of $D_0$ on the final modelling results, even when varied by orders of magnitude.

**Table 3: Symbols used in the luminescence surface exposure dating model.**

| Symbol | Unit | Description |
|---|---|---|
| $L$ | dimensionless | Luminescence signal |
| $x$ | mm | Depth |
| $t$ | a | Exposure time |
| $r'$ | dimensionless | Recombination centre distance |
| | Electron trapping | |

| $\dot{D}$ | Gy a$^{-1}$ | Environmental dose rate |
|---|---|---|
| $D_0$ | Gy | Characteristic dose of saturation |
| | Optical electron detrapping | |
| $\overline{\sigma\varphi_0}$ | a$^{-1}$ | Decay rate |
| $\sigma$ | mm$^2$ | Photoionisation cross section |
| $\varphi_0$ | mm$^{-2}$ a$^{-1}$ | Photon flux |
| $\mu$ | mm$^{-1}$ | Light attenuation coefficient |
| | Athermal electron detrapping | |
| $s$ | s$^{-1}$ | Frequency factor |
| $\rho'$ | dimensionless | Recombination centre density |
| | Erosion | |
| $\dot{\varepsilon}$ | mm a$^{-1}$ | Surface erosion rate |

The term describing optical electron detrapping contains two unknown parameters- $\overline{\sigma\varphi_0}$ and $\mu$- which have been shown to vary greatly across different lithologies, minerals and locations (e.g. Sohbati et al., 2012; Lehmann et al., 2018; Ou et al., 2018). Constraining the values of $\overline{\sigma\varphi_0}$ and $\mu$ is one of the biggest challenges in luminescence surface exposure dating. One method of doing so is by calibration from the luminescence profiles of independently known exposure age samples, provided that the calibration and unknown age samples are from the same region and preferably of similar mineralogical composition and orientation (Meyer et al., 2018; Gliganic et al., 2019; Fuhrmann et al., 2022). Previous calibrations have involved the use of historical records (Lehmann et al., 2018), road cut outcrops (Sohbati et al., 2012; Smedley et al., 2021) or the creation of a freshly exposed surface that can be resampled at a later date (Gliganic et al., 2019). In this study, we created sample specific calibration samples by exposing fresh surfaces for ~1 year at each sample site, and subsequently exploited the luminescence signal formed within this year of exposure to calculate the unknown $\overline{\sigma\varphi_0}$ and $\mu$ values for all three lithologies and luminescence signals. At one site, we were able to collect calibration samples in two different orientations to improve our understanding regarding the influence of orientation on a luminescence profile with depth (Supplementary Materials).

The data were inverted using a Monte Carlo approach to constrain $\overline{\sigma\varphi_0}$, $\mu$ and $t$. Since each unknown age sample has a site-specific calibration sample, the calibration and unknown age samples were solved for simultaneously using the same $\overline{\sigma\varphi_0}$ and $\mu$ values. To do this, for each sample, at first a luminescence profile with depth for the unknown age surface was generated using random values of $\overline{\sigma\varphi_0}$, $\mu$ and $t$, and compared to the observed measured profile using a misfit function as follows:

$$\text{misfit}_{unknown} = \sum_{i=1}^{n} \frac{1}{a} \left| \left(\frac{Ln}{Tn}\right)_{meas}^{(i)} - \left(\frac{Ln}{Tn}\right)_{pred}^{(i)} \right| \tag{2}$$

where $n$ is the number of rock slices in a sample, $a$ is the standard deviation of the plateau in the luminescence depth profile determined qualitatively, and $\frac{Ln}{Tn}$ is the luminescence signal for each rock slice where $\left(\frac{Ln}{Tn}\right)_{meas}^{(i)}$ represents the luminescence signal measured in the sample and $\left(\frac{Ln}{Tn}\right)_{pred}^{(i)}$ is the luminescence signal predicted using the random parameter values and Equation 1.

This misfit calculation was then also done for the known-age calibration surface with the same values of $\overline{\sigma\varphi_0}$ and $\mu$, although the exact number of days of exposure ($\approx$1 year) was used instead of the randomly generated $t$ value

applied for the predicted unknown age profile. Next, the sum of the misfits (misfit$_{combined}$) generated from the two profiles was used to estimate a likelihood value using:

$$\mathcal{L} = \exp\left(-\frac{1}{2}\text{misfit}_{combined}\right) \tag{3}$$

Finally, a rejection algorithm of likelihood < R was applied, where R was a randomly generated value between 0 and 1. A probability density function of $\overline{\sigma\varphi_0}$, $\mu$ and $t$ was constructed from the values that were retained. To ensure that the parameter space was sufficiently explored, we ran $1.25\text{x }10^8$ trials during the Monte Carlo search for each individual sample, with $\overline{\sigma\varphi_0}$ values between $10^{-7}$ and $10^{-5}$ s$^{-1}$, $\mu$ between 1 and 3 mm$^{-1}$ and $t$ between 1 and 200 years.

### 2.3 Estimating erosion rates

Since the $^{10}$Be concentrations and OSL depth profiles in a rock surface are both influenced by exposure and surface erosion, an erosion history can be inferred by jointly inverting the $^{10}$Be and OSL data, as described in Lehmann et al. (2019). While the effects of complex, stochastic erosion histories have been investigated (Brown and Moon, 2019), here we assume a simple, step wise erosion history where, at a specific time in the past, the surface goes from experiencing no erosion to an instantaneous onset of fixed rate of erosion. The inversion method tests random pairs of erosion ($\dot{\varepsilon}$) and erosion onset times ($t_s$) in log space to find the pairs representative of erosion histories which are most likely to successfully reproduce the $^{10}$Be and OSL data measured from the bedrock surfaces. In this study, we tested $10^4$ pairs of $\dot{\varepsilon}$ and $t_s$ with a range of possible $\dot{\varepsilon}$ values from $10^{-6}$ to $10^{-2}$ mm a$^{-1}$ and $t_s$ values from $10^{-1}$ to $10^4$ a. The $\overline{\sigma\varphi_0}$ and $\mu$ values used in the inversion were the median values found in Tables S2, S3 and S4.

**Table 1: Summary of samples and measurements. All errors correspond to 1σ and encompass propagated uncertainties from the AMS measurements, blank correction and the local production rate.**

| Site ID | Latitude WGS 84 | Longitude WGS 84 | Elevation (m a.s.l) | Lithology | Independent exposure age estimate [a] (a) | Surface orientation (strike/dip) | Thickness (cm) | Density (g cm$^{-3}$) | Topographic shielding factor | $^{10}$Be/$^{9}$Be blank corrected [b] ($10^{-14}$ at g$^{-1}$) | $^{10}$Be/$^{9}$Be uncertainty (%) | $^{10}$Be conc. x $10^3$ (at g$^{-1}$) | Exposure age (ka) |
|---|---|---|---|---|---|---|---|---|---|---|---|---|---|
| GG01 | 45.9884 | 7.8052 | 3251 | Hornfels | - | 075/53 S | 4 | 2.75 | 0.81 | 9.21 | 4.8 | 75.76 ± 3.65 | 1.91 ± 0.1 |
| GG02 | 45.9809 | 7.8003 | 2915 | Schist | - | 080/55 S | 4 | 3.00 | 0.84 | 47.16 | 3.2 | 388.74 ± 12.63 | 11.69 ± 0.46 |
| GG03 | 45.9797 | 7.7994 | 2828 | Gneiss | - | 100/60 S | 4 | 2.39 | 0.81 | 49.4 | 3.1 | 372.42 ± 11.55 | 12.24 ± 0.49 |
| GG04 | 45.9766 | 7.8003 | 2659 | Gneiss | 40 | 085/45 S | 4 | 2.52 | 0.91 | 20.02 | 3.8 | 103.22 ± 3.96 | 3.53 ± 0.17 |
| GG05 | 45.9763 | 7.8001 | 2626 | Schist | 40 | 070/61 S | 4 | 2.26 | 0.79 | 2.05 | 16.9 | 9.19 ± 1.55 | 0.35 ± 0.06 |
| GG06 | 45.9761 | 7.8005 | 2610 | Gneiss | 22 | 085/45 S | 4 | 2.32 | 0.91 | 8.92 | 6.5 | 88.24 ± 5.71 | 3.11 ± 0.22 |

[a] estimated using old geological maps and aerial photos provided by the Swiss Federal Office of Topography.
[b] $^{10}$Be/$^{9}$Be blank ratio used for the correction is: $3.2 \pm 1.7 \times 10^{-15}$.

**Table 4: Summary of inferred erosion histories across the three luminescence signals.**

| Site ID | IRSL$_{50}$ Erosion rate (mm a$^{-1}$) | IRSL$_{50}$ Minimum erosion onset time (a) | OSL$_{125}$ Erosion rate (mm a$^{-1}$) | OSL$_{125}$ Minimum erosion onset time (a) | post-IR IRSL$_{225}$ Erosion rate (mm a$^{-1}$) | post-IR IRSL$_{225}$ Minimum erosion onset time (a) | Average erosion rate (mm a$^{-1}$) | ± 1σ |
|---|---|---|---|---|---|---|---|---|
| GG01 | 3.43E$^{-2}$ | 192 | 3.13E$^{-2}$ | 152 | 7.22E$^{-2}$ | 95 | 4.59E$^{-2}$ | 0.02 |
| GG02 | 1.83E$^{-1}$ | 95 | Transient state | | 1.12E$^{-2}$ | 690 | 9.72E$^{-2}$ | 0.09 |
| GG03 | Transient state | | 1.15E$^{-1}$ | 76 | Transient state | | 1.15E$^{-1}$ | 0 |
| GG04 | Transient state | | Transient state | | Transient state | | | |
| GG05 | 2.41E$^{-1}$ | 17 | 1.52E$^{-1}$ | 15 | 6.00E$^{-2}$ | 19 | 1.51E$^{-1}$ | 0.07 |
| GG06 | 4.97E$^{-2}$ | 22 | 1.83E$^{-1}$ | 19 | 1.59E$^{-1}$ | 17 | 1.32E$^{-1}$ | 0.06 |

## 3. Results

### 3.1 [10]Be ages

The [10]Be apparent exposure ages, assuming zero correction for erosion, showed no trend with elevation (Table 1). The highest elevation sample (GG01) is younger than suggested from geomorphic ice thickness reconstructions (Bini et al., 2009) which could reflect periglacial, rather than glacial, exposure (Gallach et al., 2018, 2020). However, post-glacial erosion of the surface cannot be eliminated entirely as there exists large uncertainties in LGM ice thickness reconstructions, due to discrepancies in results derived from geomorphological observations and models/simulations. These differences are up to 800 m in some areas of the European Alps (Becker et al., 2016, 2017). On the other hand, samples GG02 and GG03 yield ages commensurate with the decay of the Egesen stadial glaciers, which has been dated to 13.0-11.5 ka (e.g. Ivy Ochs et al., 2009; Protin et al., 2019) and these results are in agreement with a set of exposure ages calculated from polished bedrock samples at the nearby Triftje glacier (Kronig et al., 2017). This provides valuable information on Younger Dryas ice thicknesses in the European Alps, and interestingly is similar to findings in Lehmann et al. (2020) nearby. The [10]Be ages for the three lower elevation samples (GG04, GG05 and GG06) of 0.35-3.52 ka contrast with information obtained from the Swiss Federal Office of Topography's old geological maps and aerial photos, which show these surfaces were only exposed 22-40 years ago. This implies that the samples suffer from inheritance, which is noteworthy because it would mean that that the Gorner glacier advanced at one point during the Holocene but did not erode the ~ 3 metres necessary to reset the [10]Be signal, as one may expect. The occurrence of inheritance in the three lower elevation samples reveals the complicated exposure history these surfaces have experienced, reinforced by studies across the European Alps which imply repeated oscillations in glacier extent during the Holocene following the Egesen stadial (e.g. Hormes et al, 2001; Ivy-Ochs et al., 2009; Goehring et al., 2011; Kronig et al., 2017; Protin et al., 2019). While glaciers in the European Alps were likely smaller than present day during the middle Holocene (e.g. Solomina et al., 2015), evidence for subsequent re-advances in glacier extent at the Gorner glacier during the Löbben period and Little Ice Age has been determined (Holzhauser, 1995, 2010), as well as at various sites across the European Alps (e.g. Holzhauser, 1995, 2010; Ivy-Ochs et al., 2009; Schimmelpfennig et al., 2014; Kronig et al., 2017; Protin et al., 2019).

### 3.2 OSL unknown parameters

For all the samples, at least three cores were measured for both the unknown age and calibration samples, and visual assessment of the luminescence profiles with depth confirmed that the surfaces had recorded only one exposure event. Furthermore, as expected, all unknown age sample bleaching consistently penetrated to a deeper depth when compared to their corresponding calibration sample. The results from investigating the effect of calibration sample orientation revealed that, across the three luminescence signals, the $\overline{\sigma\varphi_0}$ and $\mu$ values from the two different orientations overlap within $2\sigma$ (Supplementary Materials). This suggests that the effects of sampling a calibration sample in a different orientation to the unknown age sample are minimal.

For all three luminescence signals, the inversion produced probability density functions of the unknown luminescence parameters, with the best suited $\overline{\sigma\varphi_0}$, $\mu$ and $t$ values for each sample summarised in Tables S2, S3 and S4. Figure 2 shows an example result of the IRSL$_{50}$ unknown parameter inversions, taken from sample GG06. Overall, the $\mu$ values ranged from 0.59 to 2.45 mm$^{-1}$, 1.24 to 2.55 mm$^{-1}$ and 1.03 to 2.66 mm$^{-1}$ for the IRSL$_{50}$,

OSL$_{125}$ and post-IR IRSL$_{225}$ signals, respectively. For the $\overline{\sigma\varphi_0}$ values, the values ranged from 9.17 x 10$^{-7}$ to 1.82 x 10$^{-6}$ s$^{-1}$ for IRSL$_{50}$, 1.33 x 10$^{-7}$ to 1.50 x 10$^{-6}$ s$^{-1}$ for OSL$_{125}$ and 1.07 x 10$^{-7}$ to 7.34 x 10$^{-6}$ s$^{-1}$ for post-IR IRSL$_{225}$. As shown in Fig. S3, the $\overline{\sigma\varphi_0}$ solutions are all of comparable magnitude and overlap within 1$\sigma$. This is promising as the $\overline{\sigma\varphi_0}$ parameter is region and mineral dependent (Sohbati et al., 2011) and it is expected that samples from the same location share similar values. In contrast, the $\mu$ parameter results vary more than anticipated. Although

for each sample, aside from sample GG02, the $\mu$ values from the three signals all overlap within 1$\sigma$, samples of the same lithology down the transect do not have overlapping results, indicating no trend between $\mu$ and lithology for this sample set. We speculate that the observed spread reflects mineralogical variations (e.g. Meyer et al., 2018). Finally, the apparent OSL exposure ages (*t*) calculated were orders of magnitude lower than suggested by their setting and $^{10}$Be results (Tables S2, S3 and S4).

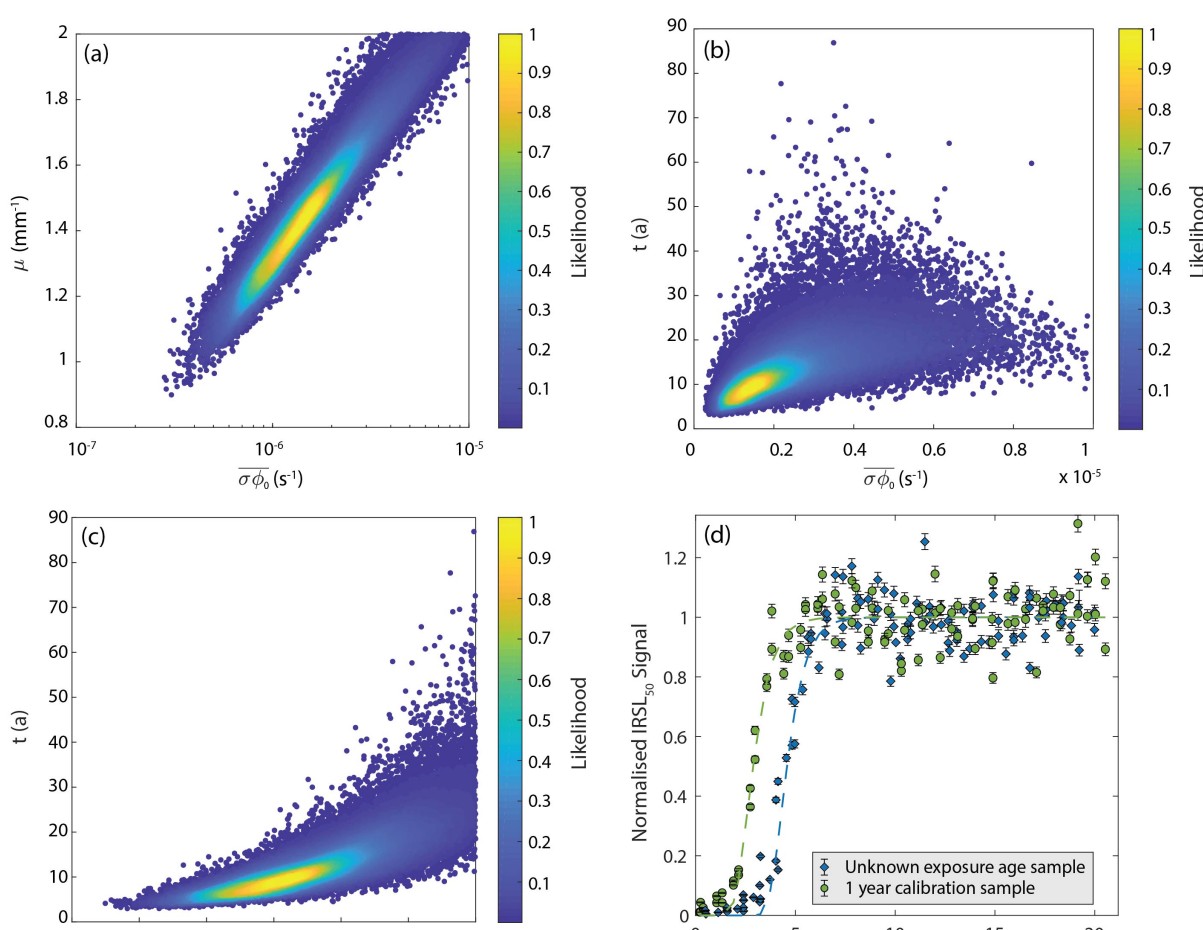

**Figure 2: Probability distribution inversion results for the unknown IRSL$_{50}$ parameters $\overline{\sigma\varphi_0}$ , $\mu$ and $t$ (a-c) and luminescence depth profiles (d) from sample GG06. In the luminescence depth profile, the blue dots represent IRSL$_{50}$ luminescence measurements for the unknown exposure age sample, and the green dots are for the known age calibration sample that was exposed for ~1 year. The dashed lines are the corresponding model fits, using the median $\overline{\sigma\varphi_0}$ and $\mu$ values and their respective exposure times. Measurement errors are derived from the square root of the luminescence counts.**

**3.3 Erosion histories**

The inversion outcomes for $\dot{\varepsilon}$ and $t_s$ for all three luminescence signals, following the method described in Sect.
2.4, are reported in Table 4. The inverted steady state erosion rates calculated across the three signals are generally consistent- an example from sample GG01 is shown in Fig. 3. Across all the samples, for the $IRSL_{50}$ signal, the rates varied from 3.43 x $10^{-2}$ mm $a^{-1}$ to 0.24 mm $a^{-1}$, $OSL_{125}$ from 3.13 x $10^{-2}$ mm $a^{-1}$ to 0.18 mm $a^{-1}$ and post-IR $IRSL_{225}$ from 1.12 x $10^{-2}$ mm $a^{-1}$ to 0.16 mm $a^{-1}$ (Table 4). Since the three lower elevation samples suffered from inheritance rendering the $^{10}$Be ages unusable, exposure age information from the historical maps and photos were employed (Table 1) for the inversion of these surfaces' post-glacier erosion rates using a slightly altered version of the inversion code. Of the six samples, the majority had luminescence profiles in steady state with erosion, thus allowing for the extraction of $\dot{\varepsilon}$ and $t_s$ values. Conversely, some samples (e.g. GG04) or signals (e.g. $OSL_{125}$ for GG02) reflected a transient state whereby a wide range of $\dot{\varepsilon}$ and $t_s$ combinations were able to explain the data, and so were therefore excluded from further analysis.

Luminescence depth profiles and probability density plots of $\dot{\varepsilon}$ and $t_s$ were generated for each sample, and the $IRSL_{50}$ results from samples GG02 and GG04 are shown as examples in Fig. 4. Each luminescence-depth plot includes the experimental data measured from the samples, as well as a reference profile (dashed black line) plotted using solely the $^{10}$Be age without correcting for erosion. As seen in Fig. 4, there is a clear mismatch between the depth of the experimental data when compared to that of the $^{10}$Be reference plot- in the case of sample GG02 it translates to a difference in depth of 12 mm. As they both record the same exposure event, this discrepancy in depth confirms the presence of surface erosion, and explains the unexpectedly low apparent OSL exposure ages mentioned in Section 3.2. Erosion removes material from the surface and therefore alters the depth of the luminescence profiles in samples so that they are shallower than what would be observed with a non-eroding profile. Previous studies (Lehmann et al., 2018, 2019; Sohbati et al., 2018; Smedley 2021) have also reported underestimations in OSL apparent ages as a result of high erosion rates (> 1 mm $ka^{-1}$). When plotting a profile using Equation 1, and the most likely solutions of $\dot{\varepsilon}$ and $t_s$ (likelihood > 0.95), it is immediately clear that these are a better fit to the experimental data (red lines).

Combining the inferred $\dot{\varepsilon}$ at steady state with the corresponding minimum $t_s$, we can calculate the minimum amount of material that has been removed from the surfaces as a result of erosion. For all three signals, the results indicate that the higher elevation samples have had more material removed than the lower elevation samples. For example, the $IRSL_{50}$ data suggest that the highest elevation samples, GG01 and GG02, have had at least 7 mm and 20 mm removed respectively, as opposed to the lower elevation samples, GG05 and GG06, which have lost 4 mm and 1 mm. This is also supported by the natural texture of the sites.

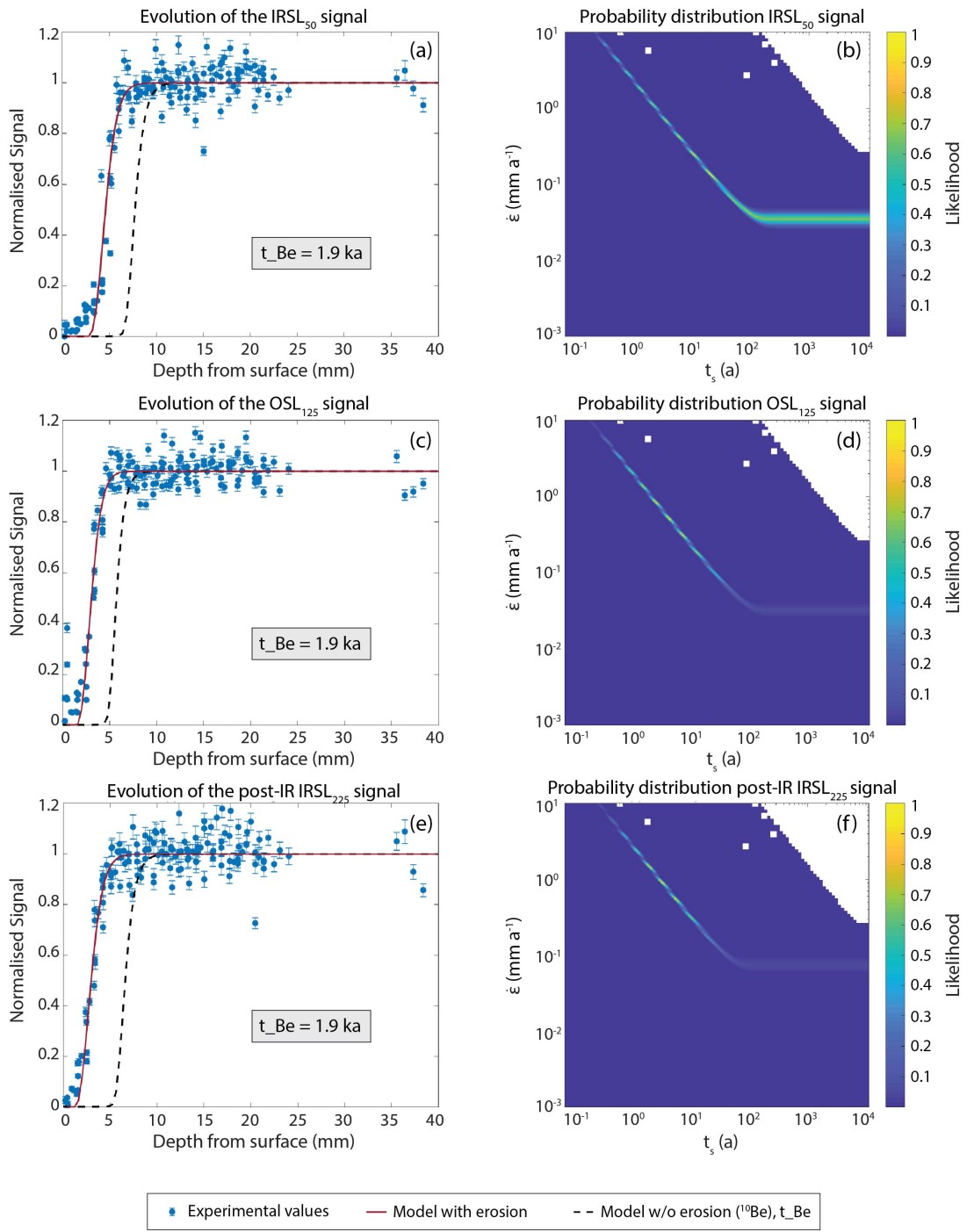

**Figure 3: Luminescence depth profiles and probability distribution inversion results for sample GG01 across the three luminescence signals used in this study- IRSL$_{50}$ (a-b), OSL$_{125}$ (c-d) and post-IR IRSL$_{225}$ (e-f)- which were inverted independently. For the luminescence profiles (a,c,e) the blue dots represent the luminescence measurements at that particular depth. Measurement errors are derived from the square root of the luminescence counts. The dashed black line represents the reference profile expected when using the [10]Be exposure age, uncorrected for erosion, and the red lines are the inverted solutions using the erosion model and the values of $\dot{\varepsilon}$ and $t_s$ deemed most likely to fit the data (likelihood > 0.95). The white zones (panels b, d and f) represent the pairs of $\dot{\varepsilon}$ and ts which cannot predict the [10]Be concentration of the sample.**

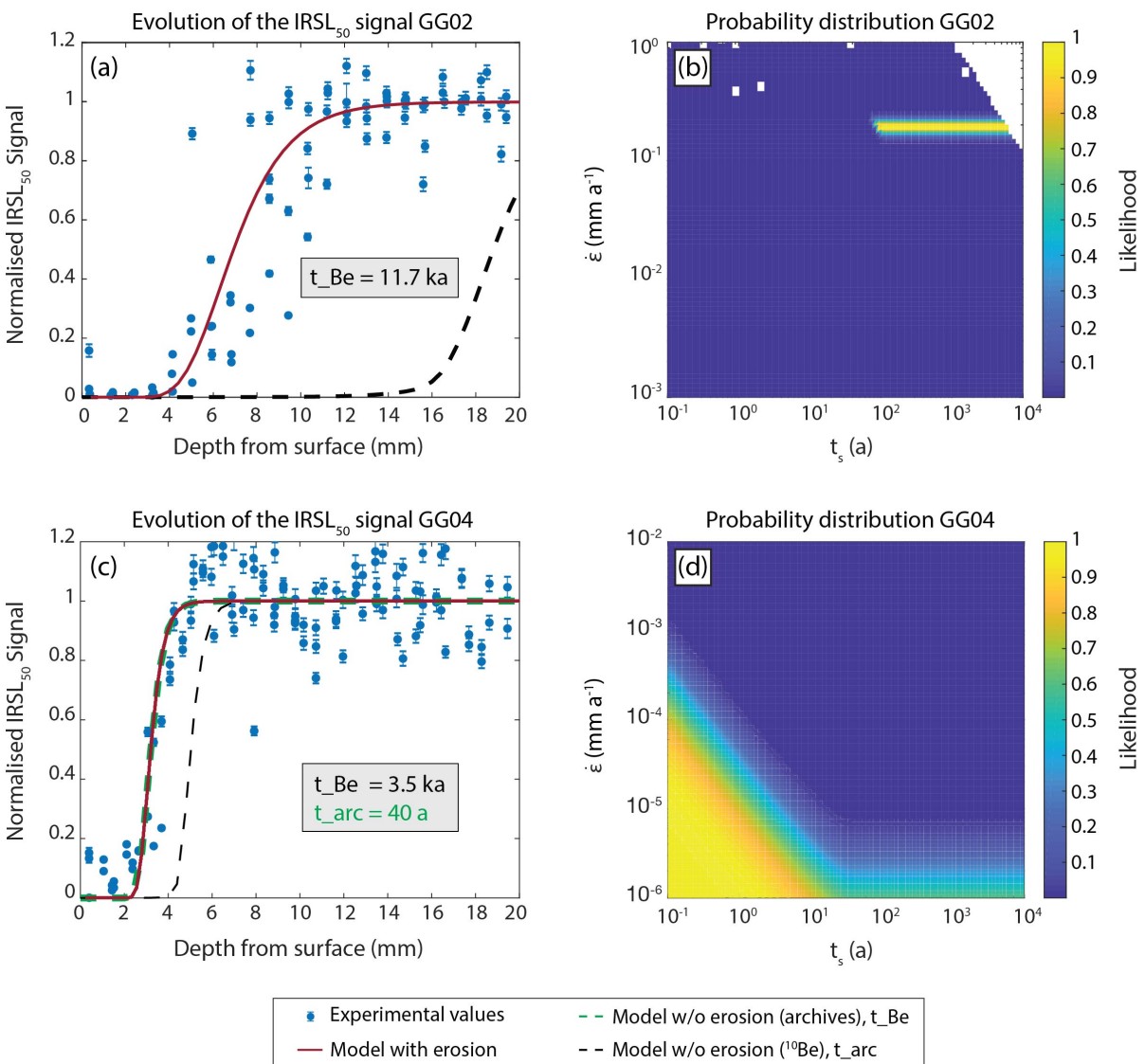

**Figure 4: IRSL₅₀ luminescence depth profiles and probability distribution inversion results for samples GG02 (a-b) and GG04 (c-d). For the luminescence profiles (a,c) the blue dots represent IRSL₅₀ luminescence measurements at that particular depth. Measurement errors are derived from the square root of the luminescence counts. The dashed black line represents the reference profile expected when using the $^{10}$Be exposure age, uncorrected for erosion, and the red lines are the inverted solutions using the erosion model and the values of $\dot{\varepsilon}$ and $t_s$ deemed most likely to fit the data (likelihood > 0.95). The luminescence depth profile for GG04 (c) also includes a reference profile (dashed green line) when using an exposure age informed from archives of old geological maps and aerial photos. This profile is overlain by the other profiles generated using the erosion model, confirming the information gleaned from the corresponding probability distribution plot that this surface has experienced very low erosion rates. The probability distributions highlight the difference between a sample in steady (b) or transient (d) state with erosion. In panel (b), the white zone represents the pairs of $\dot{\varepsilon}$ and $t_s$ which cannot predict the $^{10}$Be concentration of sample GG02.**

## 4. Discussion

### 4.1 Steady state/transient erosion histories

As mentioned in Section 3.3, while the majority of the sample and signal outputs were in steady state, some transient erosion states were also observed. In the case of sample GG04, where all three signals consistently indicate a system in a transient state of erosion, this is likely reflecting a response to a localised stochastic erosion process (e.g. surface spallation) that has removed sufficient surface material to place all the luminescence depth profiles in disequilibrium. In some cases, however, within the same sample there existed discrepancies between the three luminescence signals - some signals were in steady state while others in a transient state, a phenomenon that has also been observed by Smedley et al. (2021). Here, this was seen for the $IRSL_{50}$ and post-IR $IRSL_{225}$ signals in sample GG03 and the $OSL_{125}$ signal in sample GG02. One potential explanation lies in the luminescence bleaching depths (Figure S4). As expected, for all the samples, the $IRSL_{50}$ signal is always bleached deepest, which has also been reported in previous studies (e.g. Smedley et al., 2021; Fuhrmann et al., 2022). On the other hand, there is no clear pattern with regards to the $OSL_{125}$ and post-IR $IRSL_{225}$ signals. In some samples, the $OSL_{125}$ signal is bleached more deeply (e.g. GG05) than the post-IR $IRSL_{225}$, yet in other samples the opposite trend occurs (e.g. GG02). Since, for these samples, both signals are generally more difficult to bleach than the $IRSL_{50}$ signal, this increases their sensitivity to erosion and thus renders them prone to transient states of erosion, which could explain the presence of both transient and steady state erosion systems across signals within the same sample. Unfortunately, the reason for the transient $IRSL_{50}$ signal in sample GG03 remains unclear.

### 4.2 Dominant influences on post-glacier erosion rates

Several factors, often working in combination with each other, modulate bedrock surface erosion rates. These include, but are not limited to, temperature, elevation and surface slope. A global compilation of $^{10}$Be erosion rate measurements from bedrock surfaces, integrated over $10^3$-$10^6$ years across various tectonic settings, climate zones and lithologies, failed to find a parameter that strongly dictates outcrop erosion rates (Portenga and Bierman, 2011). This contrasted with the results from drainage basin erosion rates, where mean basin slope was revealed to be the most dominant factor (Portenga and Bierman, 2011). Lithology is known to sometimes play a dominant role in modulating the bedrock surface erosion (e.g. Twidale, 1982; Ford and Williams, 1989; Moses et al., 2014). This is because the degradation of rocks (weathering) by either chemical or physical means is what subsequently provides material for transport (erosion), and the rate of this breakdown can be primarily controlled by rock lithology. For example, studies in Northern Europe (André, 2002b; Nicholson, 2008) calculated post-glacier erosion rates using reference surfaces, and suggested that, in some cases, lithology and/or biotic influences have a greater influence on the breakdown of crystalline rocks than environmental or climatic factors. However, in this study, there does not appear to be a relationship between the erosion rates calculated and lithology of the samples (Figure S5). This could be explained by the metamorphic nature of the rocks (hornfels, schist and gneiss) that is rendering the surfaces more resistant than other lithologies. Since lithology is not a dominant influence on weathering in our study area, then the erosion rate must be controlled by other environmental or climatic factors.

To further investigate the potential influences on erosion rate in our study area, the inferred post-glacier erosion rates were plotted against elevation and surface slope (Fig. 5). When looking at the signals individually, the $OSL_{125}$ and post-IR $IRSL_{225}$ results reveal an anti-correlation between post-glacier erosion rates and elevation, whereas no trend is observed in the $IRSL_{50}$ data (Table 4). Although the luminescence signals target different minerals and

traps, they are all still from the same sample and thus have experienced the same history. Based on this, an average of the three signals was calculated for each site to generate one post-glacier erosion rate value. The trend in erosion rate and elevation observed for the $OSL_{125}$ and post-IR $IRSL_{225}$ data is maintained when analysing these erosion averages down the transect (Table 4, Fig. 5). Overall, the results exhibit a strong negative correlation between average erosion rate and elevation ($r^2 = 0.95$) but no correlation between erosion rate and surface slope ($r^2 = 0.03$).

This trend between erosion rate and elevation is in agreement with a study undertaken at the Mont Blanc Massif nearby which also found a negative correlation between erosion rate and elevation ($r^2 = 0.53$) that was stronger than the positive correlation between erosion rate and slope ($r^2 = 0.22$) (Lehmann et al., 2020).

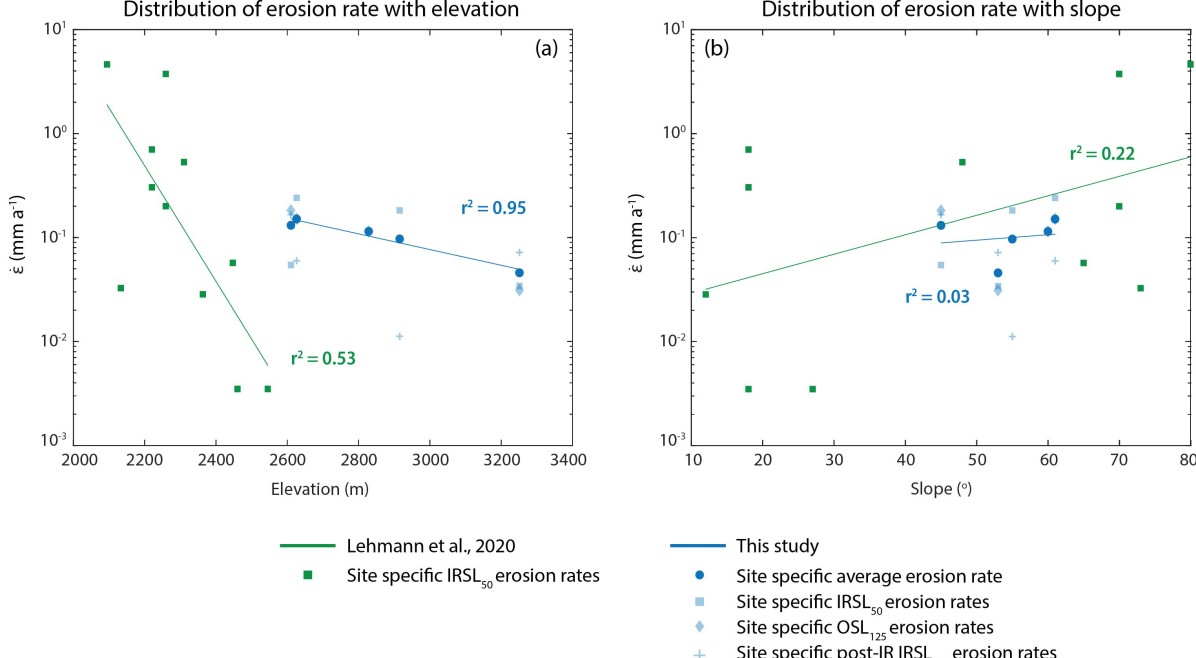

**Figure 5: Distribution of inverted bedrock surface erosion rates $\dot{\varepsilon}$ with elevation (a) and surface slope (b). The samples presented in this study are shown in blue (dark blue are the average values, light blue are the individual values from all three luminescence signals) alongside results from a nearby study at the Mont Blanc Massif in green (Lehmann et al., 2020).**

The anti-correlation between erosion rate and elevation at these two sites is surprising since surfaces at these elevations in mountain environments are typically exposed to frost crack weathering. This occurs when rocks are

subjected to temperatures between -3°C and -8°C, termed the frost crack window (Anderson., 1998), and we would therefore expect increasing erosion rates with elevation. A similar observation was made by Small et al. (1997), who found that the bedrock erosion rates from their study, located in an alpine setting, were surprisingly similar to values reported from other environments (excluding arid settings) even though frost crack weathering should be more present. While the presence of snow can help protect the bedrock by maintaining the bedrock

surfaces temperature at around 0°C, the lack of correlation between slope and erosion rate for this study site, and the weak correlation at the Mont Blanc Massif, implies that frost crack weathering is perhaps not a dominant form

of post-glacier erosion in these areas. This is further supported when visually observing the sampling sites (Fig. S1), as, aside from sample GG04, there is no clear evidence of rockfall scars or surface spallation in the other sites, suggesting that bedrock erosion is most likely occurring through continuous grain-by-grain erosion.


Potential explanations for the apparent trend in erosion rate with elevation could include: (1) wind erosion as a result of katabatic winds coming from the glacier surface (Oerlemans and Grisogono, 2002), resulting in increased exposure of surfaces near the glacier to wind erosion compared to surfaces higher up the valley sides, (2) the accumulation of water at lower elevations downslope due to gravity, facilitating erosion mechanisms that require

the presence of water, (3) increased precipitation at the lower elevation sites, following the Clausius-Clapeyron relationship which estimates a ∼25% increase in the water holding capacity of the atmosphere compared to the highest elevation site (Supplementary Materials), resulting in greater chemical weathering and subsequent erosion, or (4) observed patterns of glacial erosion in valley profiles due to quarrying and/or abrasion both scaling with ice sliding velocity (Harbor, 1992; Fabel et al., 2004; Goehring et al., 2011; Wirsig et al., 2016a; Herman et

al., 2021) and the subsequent damaging effect of the variation in ice load on the underlying bedrock (e.g. Leith et al., 2014). To further explain (4), higher glacial erosion rates were likely present at lower elevations (due to fastest ice sliding velocities) and therefore could have inflicted more damage to the underlying bedrock than surfaces at higher elevations experiencing lower ice sliding velocities. This would have weakened the lower elevation surfaces to a greater extent, rendering them more susceptible to post-glacier erosion mechanisms following ice

retreat.

Although the two study areas both observe a negative correlation, it is clear from Fig. 5 that the decrease in erosion rate with elevation is more pronounced at the Mont Blanc Massif than in this study area. This difference is likely due to local variations influencing the dominant post-glacier erosional mechanisms. While a definitive explanation

for this is still unclear, several possibilities exist. One option is differences in lithology- all the Mont Blanc Massif samples are from the same lithology (granite) and of igneous origin, whereas our sites cover three different lithologies that are all metamorphic. Alternatively, this observation could be due to elevation, since the samples in this study were collected at a higher elevation than the Mont Blanc Massif samples. Finally, the contrast in slope might be reflecting a potential relationship between erosion rate and exposure time as, due to the stochastic

nature of weathering, surfaces exposed for shorter periods of time have the potential to derive higher erosion rates than actual long-term averages (e.g. Koppes and Montgomery, 2009; Ganti et al., 2016; Lehmann et al., 2020; Smedley et al., 2021). Here, the Mont Blanc Massif samples (with the more pronounced relationship) have a greater difference in exposure times (between ∼20 ka and a few years) than the Gorner glacier samples (Table 1). Sample GG01 from the Gorner samples does not conform to this potential relationship, since it has a younger

exposure age, but this might be due to a periglacial erosional influence on its exposure age. However, even though the slopes of the trends differ, it is encouraging that the erosion rates from the two studies are comparable and that they both present a negative correlation between elevation and erosion rate.


**4.3 Comparison with other bedrock erosion studies**

Attempts to quantify bedrock surface erosion rates has been done worldwide using a variety of techniques integrated across different time scales. TCN methods are generally representative of long-term averages ($10^3$-$10^7$ years) (Small et al., 1997; Heimsath and McGlynn, 2007; Portenga and Bierman, 2011), whereas other techniques exist that work on shorter timescales (centennial to millennial), such as comparisons to reference surfaces (André, 2002b; Nicholson, 2008), using the effective radii of curvature of glacial and landslide boulders as a proxy for erosion (Kirkbride and Bell, 2010) or OSL applications (Sohbati et al., 2018; Lehmann et al., 2019; 2020; Smedley et al., 2021).

In the western US mountain ranges, in-situ TCN $^{10}$Be and $^{26}$Al were used to estimate maximum mean bedrock erosion rates of $7.6 \times 10^{-3}$ mm a$^{-1}$ from high alpine summit surfaces that showed no evidence of past glaciations (Small et al., 1997), while studies in Northwest Scotland, using boulder radii measurements of glacial deposits, and Southern Norway, using reference quartz veins from ice scoured bedrock surfaces, calculated erosion rates of $3.3 \times 10^{-3}$ mm a$^{-1}$ (Kirkbride and Bell, 2010) and $5.5 \times 10^{-4}$ mm a$^{-1}$ (Nicholson, 2008), respectively. Furthermore, a recent study investigating landslide and glacier erratic boulder erosion rates in the Eastern Pamirs of China using OSL depth profiles found minimum steady state erosion rates of $<3.8 \times 10^{-5}$ and $1.72 \times 10^{-3}$ mm a$^{-1}$ (Sohbati et al., 2018). These rates are up to four orders of magnitude less than those presented in this study. However, our values are in general agreement with erosion rates reported from studies with climates broadly similar to our study area. This includes results from the Nepal high Himalayas, a study which measured bedrock TCN $^{10}$Be and $^{26}$Al in valley ridge crests and sidewalls and reported erosion rates of $8 \times 10^{-2} - 0.2$ mm a$^{-1}$ (Heismath and McGlynn, 2007). In Europe, André (2002b) used quartz veins, quartzite layers and microcline phenocrysts as reference surfaces from roches moutonnées and glaciofluvially scoured outcrops, to calculate a bedrock surface erosion rate of $0.7 - 5$ mm a$^{-1}$, and a study in the Mont Blanc Massif, applying the same technique as this study to previously glaciated bedrock surfaces, found post-glacial erosion rates of $3.53 \times 10^{-3} - 4.3$ mm a$^{-1}$ (Lehmann et al., 2019, 2020). Aside from these periglacial studies, Smedley et al. (2021) combined $^{10}$Be measurements in sandstone boulders from a rock avalanche with three luminescence signals (IRSL$_{50}$, post-IR IRSL$_{150}$ and post-IR IRSL$_{225}$) to determine interglacial erosion rates in NW Scotland over the last $\sim 4.5$ kyr. While the erosion rates derived were in a transient state, they were still comparable to the erosion rates calculated in this study, although it must be noted that only the lower range of values (6-14 mm a$^{-1}$) from Smedley et al. (2021) can be realistically maintained over the timescale investigated. In addition, our results are in agreement with a compilation of $^{10}$Be bedrock measurements which calculated an average global outcrop erosion rate of $1.2 \times 10^{-2}$ mm a$^{-1}$ (Portenga and Bierman, 2011).

Surprisingly, when comparing the results of this study, and previously calculated periglacial erosion rates, to estimations of subglacial erosion rates, the results demonstrate comparable orders of magnitude. The notion that subglacial and periglacial erosion rates are more similar than previously thought has been suggested previously (O'Farrell et al., 2009; Guillon et al., 2015). A summary of worldwide subglacial and periglacial erosion rates is displayed in Fig. 6, although readers should remain cautious that the studies in this compilation are integrated over different timescales depending on the method itself which could introduce bias in settings where erosion is stochastic.

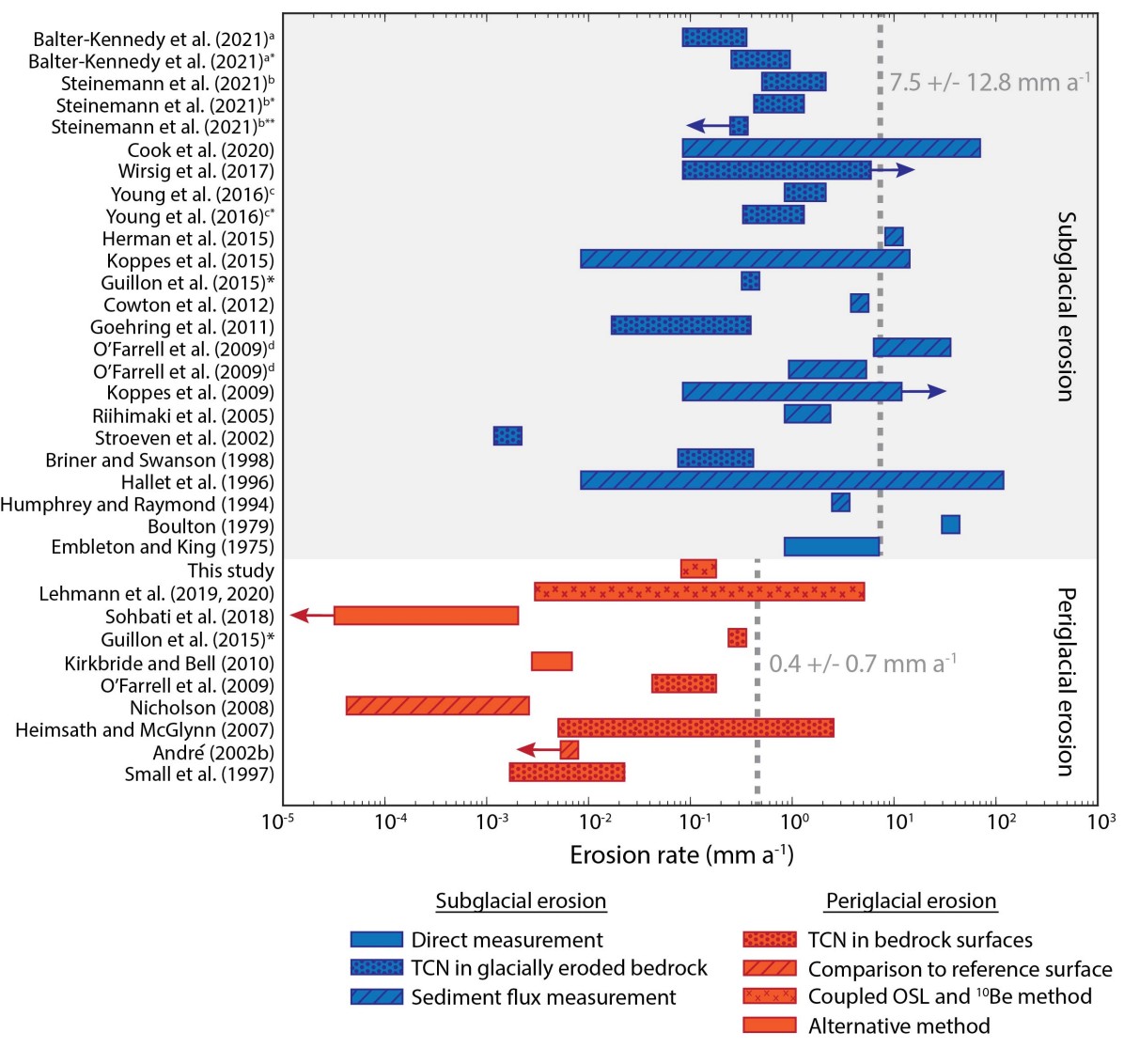

**Figure 6: Summary of the results of studies worldwide which have attempted to calculate subglacial erosion rates (blue) and periglacial erosion rates (red). Results involving bedrock of sedimentary lithology were excluded as they are not comparable to our study area. The arrows signify "greater than" or "less than". The grey dashed lines are the mean subglacial and periglacial erosion rates, accompanied by the respective mean and standard deviation values.**

Erosion at a glacier bed is primarily executed through abrasion and plucking. Although the latter is thought to be more dominant, it remains difficult to uncouple the two processes with certainty when estimating erosion rates beneath a glacier. Calculations so far have returned rates that differ by several orders of magnitude (Hallet et al., 1996; Koppes and Montgomery, 2009, 2015; Herman et al., 2015, 2021), mostly resulting from the sliding velocity of glaciers. It must be noted that the various methods are not only influenced by integration over different timescales, but for previously glaciated surfaces, also on the duration of ice cover in the corresponding study areas. In the European Alps, direct measurements from quartz veins at a glacier snout in the Swiss Alps presented

**Subglacial erosion**
- Direct measurement
- TCN in glacially eroded bedrock
- Sediment flux measurement

**Periglacial erosion**
- TCN in bedrock surfaces
- Comparison to reference surface
- Coupled OSL and $^{10}$Be method
- Alternative method

- - - - Subglacial and periglacial erosion mean values

[a] centennial timescale using $^{10}$Be of surficial bedrock surfaces and [a*] orbital timescale using $^{10}$Be depth profiles in a bedrock core.
[b] measured at marginal locations using $^{10}$Be and $^{14}$C and at riegel formations using [b*] $^{10}$Be and [b**] $^{14}$C.
[c] measured on abraded bedrock using $^{10}$Be and then [c*] calculated an approximate basin wide erosion rate (incorporating both abrasion and quarrying).
[d] reported results from two separate glaciers
*Guillon et al. (2015) applied a TCN technique but in sediment flux measurements, as opposed to bedrock.

abrasion rates of 0.9 – 3.75 mm a$^{-1}$ (Embleton and King., 1975), while measurements on marble plates cemented

to the glacier bed of the Glacier d'Argentière, France, gave a rate of 36 mm a$^{-1}$ (Boulton, 1979).

Bedrock subglacial rates have also been determined for formerly glaciated surfaces by exploiting the difference in half-lives between TCN $^{10}$Be and $^{14}$C in bedrock. In these studies, sampling deliberately targeted surfaces which displayed no apparent signs of quarrying, to attempt to isolate abrasion rates, and the results produced values

between 0.02 and > 5 mm a$^{-1}$ (Goehring et al., 2011; Wirsig et al., 2016a; 2017; Steinemann et al., 2021). The large range is due to differences in sample locations- for example, in Steinemann et al. (2021) the lower erosion rates were taken from marginal positions while higher rates from the glacial trough. From a more global perspective, the application of $^{36}$Cl in Washington, USA, found subglacial erosion rates of 0.09-0.35 mm a$^{-1}$ (Briner and Swanson, 1998) and bedrock TCN measurements in Greenland using $^{10}$Be yielded rates of 0.39-1.1

mm a$^{-1}$ (Young et al., 2016). In the study done by Young et al. (2016), the authors suggest that their results predominantly reflect subglacial abrasion, due to their sampling strategy, and proceed to estimate a likely total basin wide erosion rate of 1 – 1.8 mm a$^{-1}$ by assuming that 30-60 % of a glacier's bedrock erosion budget is attributed to quarrying. An advantage of applying TCN based measurements to calculate subglacial erosion rates is that this allows for the calculation of erosion rates at multiple points, which could subsequently reveal any

potential spatial variability in subglacial erosion rates.

Alternatively, in presently glaciated areas, contemporary sediment volume measurements at, or beyond, a glacier terminus can be coupled with ice velocities to provide insight into glacial erosion rates on a greater catchment scale. In the European Alps, studies applying this have produced values between 0.1 and 1 mm a$^{-1}$ (Hallet et al.,

1996), but findings around the globe have occasionally reported higher values (e.g. Koppes and Montgomery, 2009, 2015; Cook et al., 2020). For example, in New Zealand, glacier sliding velocities were mapped using remote sensing and combined with sediment flux measurements over 5 months at a glacier front to produce a glacial erosion value of ~10 mm a$^{-1}$ (Herman et al., 2015) while suspended sediment load measurements collected at the Leverett glacier in Greenland over two melt seasons in 2009-2010 produced a subglacial erosion rate of 4.6 $\pm$ 2.6

mm a$^{-1}$ (Cownton et al., 2012). A full compilation of glacier erosion rates, calculations and methods can be found in Herman et al. (2021). Results from sediment yield studies, and the ensuing interpretations of subglacial erosion rates, should be treated with caution as there are elements of the method which introduce potential bias. Nevertheless, the challenge of accessibility to the ice-bed interface beneath a glacier renders it difficult to estimate subglacial erosion rates by other means, and sediment flux measurements are often the only data available.


## 5. Conclusion

This study demonstrates the value of combining $^{10}$Be and OSL surface dating techniques for quantifying post-glacier bedrock erosion rate histories across time scales on the order of $10^1$ to $10^4$ years. We extended the method

introduced by Lehmann et al. (2019, 2020) by measuring three OSL signals (IRSL$_{50}$, post-IR IRSL$_{225}$ and OSL$_{125}$) for the samples in this study. The results show that using multiple OSL signals can yield, not only additional constraints for the method, but also provide information in the absence of other data- for example, the IRSL signals in sample GG03 were not in steady state with erosion, and therefore could not be used to calculate an erosion rate, but the OSL$_{125}$ signal could be used.


Averaging the erosion rate results for the three signals at each sample site resulted in post-glacier erosion rates that vary from $9.72 \times 10^{-2}$ to $1.51 \times 10^{-1}$ mm $a^{-1}$. The magnitude of the erosion rates found here at the Gorner glacier are in agreement with a nearby study at the Mont Blanc Massif (Lehmann et al., 2019, 2020). Plotting the post-glacier erosion rates against elevation and surface slope for the samples in this study indicates a strong anti-


correlation of erosion rate with elevation, and no correlation between erosion rate and slope. This is in broad agreement with the results from the Mont Blanc Massif, however the trends there are more pronounced. We suspect this is a result of local differences, such as lithology and/or elevation, influencing the dominant post-glacier erosion mechanisms present, or the reflection of a potential relationship between erosion rate and exposure time. Finally, a global compilation of both subglacial and periglacial erosion rates, reveals rates that are more


comparable than previously assumed, although subglacial erosion rates remain higher. Nevertheless, this could lead to important implications for landscape evolution models and assessing the impact of Quaternary climate on mountain erosion.

**Code availability**


The code used in this paper is available here: https://github.com/BenjaminLehmann/esurf2019.git.

**Author contributions**

FH and JE developed the project. JE organised and undertook the fieldwork with assistance from FH and BL. JE


did the OSL and $^{10}$Be preparation and analysis and performed the modelling. OS, SI and MC aided in the $^{10}$Be dating preparation and analysis. JE, GK and FH interpreted the data and contributed to writing the paper.

**Competing interests**


The authors declare they have no conflict of interest.

**Acknowledgements**

The authors would like to thank S. Vivero Andrade, D. Rech, C. Bouscary, A. Ballu, L. Malatesta and G. Prasicek


for fieldwork support as well as the three reviewers for the invaluable feedback.

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
