# Peer review of "Quantification of post-glacier bedrock surface erosion in the European Alps using 10Be and OSL exposure dating."

_EGUsphere, 2022_

## Author Comment (AC1)

*General comments*

*It is good to see development of this novel technique is continuing with the addition of the IRSL$_{50}$ and post-IR IRSL$_{225}$ signals from feldspar to the OSL$_{125}$ signal from quartz. It is encouraging that the three signals combined show similar trends.*

*The use of local calibration sites to address known variations in the optical electron detrapping parameters due to geographical and mineralogical variations is a good solution to a challenging problem.*

*Testing the effect of different sample aspect is a very valuable contribution. As is the comparison of the results to the world-wide compilation of glacial and nonglacial erosion rates which supports the idea that nonglacial erosion rates are not significantly different to glacial erosion rates.*

*Clarification is needed on some aspects of the manuscript.*

> We thank the reviewer for his positive feedback and for recognising the potential of our findings. We have addressed his comments, and this is detailed in the following section.

*Specific comments*

*The anti-correlation between erosion rate and elevation is intriguing and much less strong than in the Mont Blanc study. Please provide a reasoned explanation for the difference.*

> While a definitive answer for this remains speculative, we agree that it is an interesting observation. Therefore, we have provided three possible explanations for this difference: (1) lithology, (2) elevation and (3) a potential relationship between exposure time and erosion rate (lines 479-490).

*Although the $^{10}$Be data for the lower three samples are compromised by inheritance, could the authors deliberate on the inverted erosion rates for GG02 and GG03. The $^{10}$Be derived steady state erosion rates for these two samples are about 4.8E$^{-2}$ mm a$^{-1}$ and 5.8E$^{-2}$ mm a$^{-1}$. These are roughly half of the rates predicted by the inversion method. The $^{10}$Be derived steady-state erosion rates are directly related to the measured $^{10}$Be concentration in the samples and represent maximum steady-state erosion rates.*

*The higher erosion rates calculated using the inversion method used in this study are not compatible with the measured $^{10}$Be concentrations. It is not possible to get the measured $^{10}$Be concentrations with the calculated erosion rates. It is important that the authors state very clearly if the $^{10}$Be data was in fact used in the inversion, or did they derive the erosion rates simply from Eq. 1, which does not incorporate the $^{10}$Be data.*

*If the $^{10}$Be data was used, please explain how the erosion rates from the inversion method are reconciled with the measured $^{10}$Be concentrations. What was the exposure/erosion history of GG02 and GG03, especially given Figure 3b suggests that the inversion modelled erosion rate is invariant for erosion onset times ts >10$^2$ a. Is it the case that the OSL signal only records the*

*last few hundred years at the inversion method erosion rate, and prior to that time the samples were eroding at half the rate to accumulate the measured $^{10}$Be concentrations? If that is the explanation, what caused the acceleration in the erosion rate?*

We apologise that our methods sections were unclear. While we are not entirely sure where the values stated above for $^{10}$Be derived steady state erosion rates have come from, we assume they are from the standard $^{10}$Be erosion rate calculation method. This integrates over the entire exposure history of a sample, whereas here the inversion method only applies an erosion rate following an erosion onset time (which is not equal to the exposure time). Since the $^{10}$Be steady state erosion calculation is integrated over longer periods, this explains the lower erosion rates produced compared to the inversion erosion rates here which integrates over a shorter period.

*If the $^{10}$Be data was not used, explain why, and revise the title of the paper to reflect that $^{10}$Be data was not used to quantify the post-glacier erosion rates discussed in the manuscript.*

It was used and we hope we have clarified this with our explanation above.

***Specific comments by line number:***

*104 "…since TCN are formed ~50-60 cm (Lal, 1991) below the rock surface…" is incorrect. TCN are formed at the surface and down to several metres. The ~50-60 cm is the e-folding depth for common rock densities.*

Thank you for bringing this to our attention. We have amended the sentence accordingly.

*117 "…due its…" should be 'due to its'*

Thank you for your detailed reading of the manuscript. The typo has been corrected.

*303 Table S1 does not show summary for each sample. It shows data for Sample 5 (which I assume is GG05). Table S1 is not referred to in the main text. It is referred to in the Supplement. Table S1 in the main text should be Table S2, or change the labels in the Supp.*

It was indeed an oversight on our part to have this mismatch between the table numbers and the main text. We have now changed the labels in the Supplementary, and furthermore changed "sample 5" to "GG05" for consistency purposes.

*306 1.13 x 10$^{-6}$ is 1.8E$^{-6}$ in Table S2. Check the data.*

Thank you for pointing this out, it has been corrected.

*307 7.34 x 10$^{-7}$ is 7.3E$^{-6}$ in Table S2. Check the data.*

This has now been done.

*337 1.12 x 10$^{-2}$ is 7.22E$^{-2}$ in Table 4.*

Unfortunately, we do not follow the reviewer's point here.

*338   Add reference to Table 4 so the sentence ends… 0.16 mm a$^{-1}$ (Table 4).*

We have now done this.

*484   "…local differences…" This is vague. Please elaborate.*

Thank you for your suggestion. As mentioned in the "Specific comments" above, following the reviewer's comments, we have added a few sentences to elaborate that this observation could be a reflection of local differences such as lithology and/or elevation influencing the erosion mechanisms, or also due to the presence of a potential relationship between erosion rate and exposure time.

---

## Author Comment (AC2)

*Dear Authors, dear Editors,*

*Please find below my evaluation concerning the manuscript by Elkadi and co-authors entitled "Quantification of post-glacier erosion in the European Alps using [10]Be and OSL exposure dating" (manuscript egusphere-2022-43).*

*This manuscript investigates bedrock surface erosion in an alpine environment, focusing on post-glacial surfaces and combining OSL and in situ 10Be surface exposure dating. The authors targeted different samples along a formerly-glaciated topographic profile, and used a multi-signal OSL investigation to constrain bedrock erosion rates and durations. Their results show variable erosion rates between signals and samples, with an elevation relationship that they relate to periglacial erosion mechanisms (e.g. frost-cracking). They compare their results to recent study in a similar environment, and finally propose a compilation of "non-glacial" vs. glacial surface erosion rates (bedrock and boulder) from the literature that they discuss in terms of rates variability and magnitude.*

*This is an interesting manuscript, well-written and referenced. It follows the recent developments of OSL surface exposure dating and the original approach combining OSL with 10Be data to retrieve local erosion histories. In the present study, the authors used a multi-signal approach for OSL exposure dating, which is a very good illustration of the potential of luminescence techniques for quantifying exposure/erosion histories of bedrock or boulder rock surfaces. They also investigated the usefulness of artificial calibration surfaces, with short exposure time (1 year), to efficiently constrain bleaching parameters, as well as the influence of surface orientation on these parameters. Finally, they placed their results within a large-scale compilation of surface erosion rates from the literature, discussing the relative overlap between non-glacial and glacial surface erosion rates. I thus think that the present manuscript would be a very interesting contribution for Earth Surface Dynamics, bringing new evidence and quantification of bedrock surface erosion rates in post-glacier settings, and nicely complementing recent studies in this topic while raising fruitful discussion in the geomorphology community.*

*I have outlined below my questions and suggestions in a set of general and specific comments below. Most of my suggestions are concerning the presentation of information related to the OSL exposure dating and calibration/multi-signal investigation. In addition, I would think that more discussion about the actual erosion/weathering processes (physical mechanisms) would help the readers to better appreciate the discussion about compiled glacial/non-glacial erosion rates.*

> We thank the reviewer for his constructive feedback which has undoubtedly improved the paper, and for recognising the relevance of our work.

***General comments:***

*1 – In the present study, the authors refer to "post-glacier erosion" for the surface erosion rates they aim to quantify. I agree with the used term, although this is maybe too vague and can be specified already at the beginning of the manuscript. This should go along with a description of the sampled landscape/morphological features and a clear statement of the*

*adopted strategy: why targeting formerly-glaciated bedrock surfaces and not random surfaces in the catchment? how comparable would be then the output surface erosion rates between GG01 and other samples, given that GG01 has never been glaciated? What are the exact erosion mechanisms investigated there? Wind erosion, surface chemical weathering, frost cracking, a mixture of all?*

*This is not an easy question, but I feel that the readers will better appreciate the approach and outcomes if these are better clarified in the manuscript.*

> We define post-glacial erosion as the erosion experienced by a bedrock surface once ice has retreated. We have now included this definition to the manuscript (lines 57-58) along with an additional figure in the supplementary showing all the sampled surfaces for context. However, we would prefer not to go into more detail on the specific erosional mechanisms because at this stage it would be mostly hypothetical. The hope is that the findings from our study here will help contribute towards future investigations into erosion mechanisms in mountain environments.

> Additionally, we agree that it is beneficial to a reader to know why bedrock surfaces were targeted for the purposes of this study and have added a few sentences explaining this (lines 97-100).

> Finally, we respectfully disagree with the reviewer's assumption that GG01 has never been glaciated. In fact, there exists longstanding discrepancies between LGM ice thickness reconstructions based off geomorphological observations and models/simulations (up to 800 m in some areas of the European Alps (Becker et al., 2016; 2017)). We acknowledge that this was not mentioned in the previous version of the manuscript, and have now expanded on this in the text to make it clear to a reader.

*2 – OSL surface exposure dating. The multi-signal approach is really interesting and promising, however the comparison between signals could be extended and complemented in my opinion. First of all, this is not entirely clear to me why bleaching parameters would be similar between different signals, as we know from literature than bleaching of IR signals are more difficult/slower than OSL signal. I would encourage the authors to provide more discussion about this interesting result. Second, the output erosion rates differ between signals, can these be indicative of the uncertainty in erosion quantification? In their output results (Table 4), the authors provide estimated erosion rates but these are not associated to any uncertainty. Would it be possible to estimate some uncertainties from the likelihood results?*

*More importantly, how can we explain that some bleaching profiles are in steady state for a given signal and in transient state for another signal, within the same sample/core? This is really intriguing but would need I think more discussion.*

> Thank you for this thought provoking comment. Indeed, it appears that in rock surface dating, the $IRSL_{50}$ signal bleaches more easily than the $OSL_{125}$ and post-IR $IRSL_{225}$ signals. This contradicts bleaching rates in conventional luminescence dating where $OSL_{125}$ bleaches more rapidly than the $IRSL_{50}$ signal. While the exact cause of this remains unclear, it could be due to the fact that according to Ou et al. (2018) rock

types have higher attenuation coefficients at shorter wavelengths which would explain why the OSL$_{125}$ signal bleaches less readily than the IRSL signals. It would be expected that this translates into different bleaching parameters however, the manner in which the parametrisation occurs means that the unknown parameter values are not entirely independent, and a trade-off exists between $\overline{\sigma\varphi_0}$ and $\mu$.

We do agree however that it would be useful to expand the discussion so that it includes a section that attempts to explain the differences that result in transient/steady state erosion histories and we have now done this (Section 4.1). With regards to the uncertainties, there is a column in Table 4 that provides uncertainties on the average erosion rates.

*3 – Compilation of non-glacial/glacial surface erosion rates. This is a nice compilation and this questions the relative idea of efficient subglacial processes in shaping mountainous landscapes. However, I think several clarifications/information are missing to fully appreciate this compilation. First, this is unclear to me what are "non-glacial" erosion rates, since I have the impression that fluvial or landslide rates have not been included. So this is more a comparison between periglacial/hillslope erosion vs. glacial erosion, for the later the geomorphic agent being easily identified (subglacial ice or water). Second, I think that for surface erosion rates the setting/environment is also very important, i.e. one would expect different erosion rates for a bedrock/surface exposed since long time to atmospheric agents than a recently deglaciated surface, no?*

*Finally, I guess that the measurement time could be also important in the output erosion rate; have the authors tried to confront the compiled erosion rate to the measurement period?*

Thank you for the positive feedback and for noticing the contribution of this compilation to the research field. Indeed, it was unclear what exactly was being compared. We have now altered the figure so that it only includes studies corresponding to periglacial erosion, and have replaced the word "non-glacial" with "periglacial". The only mention of non-glacial that is left in the manuscript is at the beginning (line 36), where we have subsequently defined it in the context of the paragraph (lines 38-39).

With regards to the exposure and measurement time considerations mentioned by the reviewer, we agree that this is important but unfortunately this information is not always available for the studies used in the compilation. We have added a sentence in the manuscript cautioning readers about this (lines 532-534).

**Specific comments, by line number:**

*Line 1. "post-glacier erosion…". Maybe precise in title that this study investigates "bedrock surface" erosion, and is thus focusing rather on local/small-scale erosion and not large-scale landscape evolution (e.g. fluvial erosion…).*

Thank you for the suggestion and have changed the title so it reads: "Quantification of post-glacier bedrock surface erosion in the…".

*Line 14. "glacial and non-glacial". Please be more specific there, what is considered as "non-glacial" in the present study. Are these post-glacial evolution of glacial surfaces (by atmospheric erosion/weathering), periglacial processes or more generally fluvial/hillslope erosion? See also my general comment about this.*

We have now reworded the sentence and removed any mention of "non-glacial" here.

*Line 19. "in Zermatt, Switzerland". Maybe precise that this is located in the (central) European Alps.*

Following on from the reviewer's comment, we decided to remove "Zermatt, Switzerland" from this sentence, since this is mentioned later on in the paper (Study area, line 133) anyway and have instead written "to the Gorner glacier in the European Alps" (line 19).

*Line 24. "...could be equally important." I would suggest to add a sentence there for the potential implications of such result, this appears not entirely clear as presently phrased.*

Following this review process, we have changed our conclusions and instead decided that, while periglacial erosion rates could be higher than previously thought, subglacial erosion rates continue to have a greater influence on landscapes. Therefore, we have amended the final sentences in our abstract to reflect this, and expanded to say that this would result in transient periglacial erosion rates with changing ice thickness.

*Line 37. "...global feedback loop that exists...". Some references there would be needed to introduce this feedback loop.*

Thank you for your suggestion, however we are referring to the relationship that is mentioned a few lines above (lines 30-32) at the beginning of this paragraph, for which there are many references already provided. We do not find it necessary to repeat these references a second time so close to their original mention.

*Lines 42-43. "In contrast, studies exploring erosion during interglacial times have mainly investigated at catchment-wide erosion rates". I don't entirely agree with this statement, some studies have also investigated more local fluvial erosion (gorge incision etc., e.g. for the European Alps Korup and Schlunegger, 2007; Rolland et al., 2017; van den Berg et al., 2012; Valla et al., 2010) or the spatial distribution within a catchment (e.g. Fox et al., 2015 for the Alps). Maybe rephrase or add more information there.*

The reviewer is correct, we have now changed the sentence so it mentions local fluvial incision and added the relevant references.

*Line 42. "glacial erosion, bedrock surface erosion and rockfall". See my general comment about this, all terms refer to "bedrock surface erosion" but physical processes and scales differ. Please check and rephrase.*

We agree with the reviewer that the original wording of the sentence is misleading. It has been rephrased to: ".. contributions of various erosional processes remains challenging" (lines 51-52).

*Line 49. Again there, what is "post-glacier erosion". Hillslope, fluvial, or atmospheric weathering? This needs specification for your study.*

We are referring to the erosion that the surface has experienced since glacier retreat (i.e the material that has been removed), and we have added this information to the sentence (lines 57-58).

*Line 49. "six samples". Please precise what kind of samples (i assume glacially-polished bedrock or glacial morphologies like roches moutonnées no?). This is important to understand what processes are targeted.*

We understand the reviewer's point, but would prefer to keep the introduction as it is since there is more detail on the sampling sites later in the manuscript (Section 1.2).

*Line 75. "2 x 10-1".*

We have now done this.

*Line 83. "post-glacier erosion rates...". There is a good reason why targeting formerly-glaciated bedrock surfaces in the present study, but this is not really explicit in the introduction. Please consider adding one or two sentences on the adopted strategy and why targeting post-glacier surfaces rather than other bedrock surfaces randomly in the landscape.*

Thank you for the suggestion, we have expanded the manuscript (lines 97-100) to include a few sentences explaining the advantages associated with sampling these surfaces.

*Line 88. "converted into an exposure age". Add "apparent" there.*

We have done this.

*Line 95. "surface traps". Unclear whether these relates to traps at the rock surface or energetically for luminescence. Please rephrase. Also, maybe already precise the depth range at which the sun's energy is sufficient to reset the OSL signal (lines 96-99).*

Thank you for bringing this potential confusion to our attention. As per your suggestion, we have removed any mention of traps and amended the sentence so it reads: "...the sun's energy is sufficient to naturally reduce the surface luminescence signal to zero" (lines 110-111). However, with regards to the reviewer's second point, we prefer to keep the depth range information in its original position further down because it provides a direct/easy comparison between the OSL and TCN dating methods.

*Line 101. Maybe add "apparent" there too for exposure age.*

We have done this.

*Line 105. Maybe also include the recent work of Sellwood et al. 2019 and/or Sellwood and Jain 2022.*

While these are both excellent studies, we don't feel like the references fit with this sentence, which refers to slices, since they don't work with slices specifically. However, we agree that this work should be mentioned in the manuscript and so have added Sellwood et al. (2019) elsewhere (line 116).

*Line 106. "influenced by exposure". Sunlight exposure? Exposure time? Please specify.*

We thank the reviewer for detecting that this was too vague. The sentence has now been amended to emphasise that it says exposure time (line 122).

*Line 119. "in the local area". Not clear, please rephrase.*

The wording has now been changed from "local area" to "Western Alps" (line 135). We hope this is clearer.

*Line 121. "six sampling sites down a vertical transect". Same comment as line 49. The reader is missing a morphological/geomorphological description of the targeted bedrock surfaces (glacially-polished or not, glacial or periglacial features, etc.) and explanations for the adopted sampling strategy. This is really difficult to have a good understanding based on small insets in figure 1. Also, this is important I think to present the surface slopes for the different samples, etc.*

Thank you for your suggestion, but we prefer to keep this sentence as it is to avoid repetition since more detailed descriptions of the sampling sites can be found a few sentences further down (lines 151-155).

*Line 123. "aside from the highest sample". So this is important to explain that this sample is not reflecting "post-glacier" erosion, but periglacial erosion as this was never ice-covered or at least no during the LGM). Also, then what is the bedrock surface morphology for this sample (see my previous comment)?*

This is a good point from the reviewer. The Bini et al. (2009) reconstruction was done based on geomorphological observations, however as mentioned above, there exists a longstanding discrepancy between ice thickness reconstructions (up to 800 m in some areas (Becker et al., 2016; 2017)) when comparing results from geomorphological observations and model/simulation results. This means that we actually have no way of knowing for certain whether this sample was covered by ice or not. We have amended the main text to reflect this- both by adding the word "Geomorphological" to line 142 and also explaining this discrepancy (lines 308-311).

With regards to the bedrock surface morphology, this is described a few sentences later (lines 151-155) and also visible in an inset in Figure 1 and in the newly added Figure S1.

*Line 136, Figure 1. This is a nice figure, but not totally informative for the setting area. Is it possible to add the LGM ice contours on panel b? and to replicate the ice lines on panel c for clarity (for instance I cannot really tell if the three bottom samples have been lastly exposed in 1973 or 2009 based on panel b)? Pictures as inset in panel c are really small, and scale is missing? What is the source(s) of the photos showed in panel b and c? Another question, what is above the sample GG02, it looks like a small plateau or morainic ridge (Younger Dryas?)? Maybe consider adding also a topographic profile for the transect, on which you can locate the samples and ice thickness/extent from LGM to present-day.*

*One suggestion would be to add another figure (supp or main text) to show the sampled morphologies and potentially the different lithologies (rock-slice pictures?).*

Thank you for your useful suggestions that have improved the figure and contributed greatly towards the manuscript. We have now made the following changes:

1) Past glacier extent ice lines have been added to panel c. We hope this clarifies the exposure ages for the bottom three samples.
2) A scale has been incorporated into the insets, however we have decided against increasing their size as this would completely cover the rest of the background image. Readers can look to the newly added Figure S1 in the Supplementary if they wish to see larger images.
3) The source of panel b has been included in the caption. Panel c's source (Google Earth) is already located in the bottom left hand side of the image.
4) Additional figures showing images of (1) the sampling sites and (2) slices have been attached to the Supplementary Materials (Figures S1 and S2 respectively).

However, we have decided against adding any information related to LGM thickness (i.e. ice contours and topographic transect) as there is still debate on LGM ice thickness-particularly discrepancies between model results and geomorphological observations. Furthermore, we are unsure as to the definition of what lies above Sample GG02, it looks more like rockfall rather than a small plateau or morainic ridge, and so we prefer not to comment on this.

*Line 136. "Sample preparation". Please specify where sample preparation and chemical Be extraction have been performed.*

We have specified that the sample preparation and chemical Be extraction was performed at ETH Zurich, Switzerland.

*Line 165. There is a ) to be removed for the blank value.*

The ) in question is closing the bracket that begins before "full chemistry long-term.." (line 182) a few words back in the sentence.

*Line 189. "with a DASH head". I would suggest to describe the different filters listed in Table 2 for non-specialists.*

Thank you for your suggestion. We have included a sentence in manuscript explaining that different filters are required because the quartz and feldspar luminescence signals emit in different wavelength (lines 210-211). Furthermore, we have provided a column in Table 2 showing the different emission wavelengths of the minerals.

*Lines 192-194. Are the different criteria arbitrary or common for rock-slice luminescence? Maybe refer to technical paper to support these, e.g. Elkadi et al., 2021?*

They are arbitrary values that have also been previously used in rock slice luminescence. Following your recommendation, we have now referenced the Elkadi et al. (2021) paper for support.

*Line 205. This is unclear and not explained in the main text how equation 1 is treated with respect to the recombination distance r'. For non-specialist readers this will appear relatively obscure, given that athermal detrapping parameters are not presented for these measurements/samples. This is also similar for the dose rate parameters ($D_0$ and $\dot{D}$), no information about their values (and how $D_0$ is obtained) is provided, only description in Table 3.*

The reviewer has correctly pointed out that the manuscript was missing some key information. We have now included the $D_0$ value used (500 Gy for each sample) and elaborated that this value was chosen following sensitivity tests that revealed the negligible effect of $D_0$ on the final result, even when varied by orders of magnitude (lines 236-238). Furthermore, the $\dot{D}$ values of each sample have been added in a column in Table 1.

However, we prefer to keep the rest of this section as it is to not overwhelm a non-specialist reader, and feel it is sufficient to cite Lehmann et al. (2019) for the readers seeking further information.

*Line 220. "Previous calibration sources". Unclear, please rephrase.*

Thank you for bringing this to our attention, we have removed the word "sources" so that the phrase reads "Previous calibrations…".

*Line 224. "unknown parameter values". Please specify this parameter for clarity ($\sigma\varphi$?).*

Here, we were referring to both parameters-$\overline{\sigma\varphi_0}$ and µ. Thank you for flagging that this is unclear, we have specified this in the sentence so instead it reads: "…the unknown $\overline{\sigma\varphi_0}$ and µ values" (line 252).

*Line 225. "the influence of the surface orientation". Additionally, I think discussion about the outcome results would be interesting for readers if reported in main text, not in supp (at least briefly).*

Thank you for your interest in our results. We have added a few sentences in the results section of main text briefly explaining the outcome of the different calibration surface orientations (lines 336-339).

*Line 239. "… using the random parameter values and Equation 1".*

We have done this.

*Lines 250-251. "1.25x10^8 trials". For each individual sample or in total?*

Thank you for bringing this lack of clarity to our attention. This value is for each individual sample, and we have modified the sentence to reflect this (line 280).

*Also, I don't fully understand how the ranges for the inverted parameters have been defined, especially for exposure time t only between 1 and 200 years but setting information suggest much longer exposure times for high-elevation samples no? Please clarify on which basis/information the parameter ranges have been defined.*

The ranges were chosen after running various tests. The exposure time range in particular was set between 1 and 200 years because the highest apparent exposure age result generated during these preliminary model runs was ~ 110 years (sample GG03). We were also surprised by this low value, considering the setting information, but this can be explained by the presence of surface erosion. We hope this has clarified matters for you.

*Lines 258-259. "…simple, step wise erosion history where, at a specific time in the past, the surface goes from experiencing no erosion to an instantaneous onset of fixed rate of erosion". I am wondering whether this is possible to also have a simpler scenario where you estimate erosion rate since the exposure of the bedrock surface (i.e. ts = t from 10Be data). Have you tested this and if yes is there any compatible scenario(s) with OSL/10Be data?*

This is an interesting point from the reviewer that unfortunately we have not tested. We will not address it here as it is beyond the scope of research but it is worth further investigation in future studies.

*Line 259. For the inversion of erosion history, what are the bleaching (σφ and μ) parameters and exposure times used? Best-fitting values for bleaching parameters (Table S1)? Please clarify.*

This is a good point. Indeed, the bleaching parameters used are those in Table S1 and they are actually the median value calculated from the retained values following the rejection algorithm outlined in Section 2.2. We have added both a sentence in the main text saying this (line 293) as well as an explanatory sentence in the table captions (S2, S3 and S4).

*Line 265. Several questions for Table 1: Can you add more information for surface orientation? Two values are given, but no unit nor details. Please also provide [10]Be/[P]Be ratios in the table, so that 10Be concentrations can be recalculated in the future. Are the uncertainties reported for exposure ages internal or external?*

Thank you for your suggestions. We have now included in the surface orientation column heading that the values represent strike and dip measurements, as well as a

rough dip direction for all the samples (which is South). A column for the $^{10}$Be/$^{9}$Be ratios is now present too. With regards to your question on the uncertainties reported, they are external which we believe should be clear enough to a reader since the caption includes the following sentence: "All errors correspond to 1σ and encompass propagated uncertainties from the AMS measurements, blank correction and the local production rate."

*Line 274. "3. Results and interpretation". The presented results are already quite interpreted in this section, so I would suggest to rephrase the section label.*

Thank you for your suggestion, and while we understand your point of view, we prefer to leave the section heading as it is since we believe it would be confusing to have a "discussion" section right after "results and interpretation". However, since submitting the original manuscript, we have removed the elevation vs erosion interpretation from this section down to the discussion. We hope this further justifies maintaining the heading title as "Results".

*Line 276. "apparent exposure ages". I would also suggest to add a figure with 10Be apparent exposure ages and topography for illustration.*

Thank you for your comment, however we respectfully disagree and believe that the $^{10}$Be apparent exposure ages and sampling site elevations in Table 1, combined with the inset in Figure 1, is sufficient information for the reader.

*Line 277. "The highest elevation sample (GG01) is younger than suggested from ice thickness reconstructions (Bini et al., 2009)". If this sample has been collected above the LGM ice surface, then it reflects periglacial exposure and its apparent exposure age is not related to LGM glaciation, see for instance results in Gallach et al. 2018; 2020. Please consider rephrasing or clarifying this sentence.*

We have added a few words explaining that this could be due to periglacial erosion, as well as the references suggested by the reviewer (line 308). However, as mentioned above, there are still large uncertainties regarding LGM ice thickness, exacerbated by the longstanding mismatch that exists between geomorphic evidence and modelling (Becker et al., 2016; 2017). This means that we cannot entirely exclude the idea that the surface has experienced post glacial erosion and we have explained this (lines 308-311).

*Line 279. This is a very interesting result as you can reconstruct the YD ice thickness from your 10Be apparent exposure ages, which may be linked to this small plateau/surface just above sample GG02. Please consider expanding this result, this is relatively similar outcomes compared to Lehmann et al. 2020.*

Thank you for the suggestion and positive feedback regarding our results. Following this comment, we have expanded the sentence to include this information (lines 315-316).

*Line 288. Maybe also consider citing the work of Goehring et al. 2011 on the Rhone glacier.*

Indeed, it is a relevant reference and we have added it.

*Lines 297-299. This sentence may be moved to methods.*

While we understand the reviewer's point of view and thank him for his suggestion, we prefer to leave this sentence here as we believe the conclusions drawn from the visual assessment of the cores constitutes as a result. Furthermore, the methods section already mentions that multiple cores were taken per sample.

*Line 303. "results for each sample summarised in Table S1". I would strongly encourage the authors to present results as figures (like figure 2) for all samples, either in main text or in supplementary. This would be important for the readers to evaluate the noise in data and reproducibility between cores for each sample (old and calibration, and also for different orientations).*

We recognise why the reviewer has recommended this, but we have decided not to do this as we believe the information included in Tables S2, S3 and S4 is sufficient.

*Line 308. Is it possible to present there quickly the results about different orientations? I guess this would be interesting for some readers to have such information, not all in supplementary.*

We agree that it is useful for the readers to have some information on these results in the main text, and so have included a few sentences on the matter in lines 336-339.

*Line 313. "...mineralogical variations". Is there a link between μ values and lithology? Can the authors provide some pictures of the rock slices, especially for GG02 which seems different from others?*

Unfortunately, our sample set shows no link between μ values and lithology. We have expanded on a previous sentence in the manuscript to emphasise this observation (lines 351-352).

We thank the reviewer for recommending we include pictures of our rock slices, it is certainly helpful for future readers. These can now be found in Figure S2.

*Line 315, Figure 2. I would suggest to have at least one figure showing the bleaching profiles of the different signals, at present only IRSL50 signals are shown. Is it possible to provide such information?*

We thank the reviewer for his idea, and we agree that it is important to show the bleaching profiles of the different signals. However, we don't think this information belongs in Figure 2 where the focus is a result of the unknown parameters inversion. Instead, we have added a figure to the supplementary (Figure S4) showing the different bleaching depths of each luminescence signal in each sample as well as information in the main text regarding the relative bleaching depths between the signals (lines 406-410).

*On figure 2, inversion outcomes for t, the OSL apparent exposure time, is shown. However, this outcome is not presented in Table S2, nor discussed in the main text. I think this is important to show this, and to clearly present the differences in apparent exposure ages between OSL and 10Be data for all samples.*

Thank you for your suggestion. We have included a column in the supplementary (Tables S2, S3 and S4) containing the inversion outcomes for $t$, and added a few sentences to the main text expressing that the OSL apparent exposure times are lower than expected considering their setting and corresponding [10]Be results (lines 353-354), and elaborating on why this is the case (lines 377-380).

*Line 318. "inversion outcomes for e and ts". Please provide the range for these parameters.*

We have now done this.

*Line 320. "exposure age information from the historical maps and photos were employed". Where can the reader access the used exposure ages for these samples? Please specify in main text what exposure durations you used.*

This is a good point raised by the reviewer, thank you. There is now a column in Table 1 with the exposure ages that were used and the table is referenced at the end of the relevant sentence (line 365) for the reader to refer to.

*Line 323. "transient state". This is not totally clear what is transient state from looking at figure 3d, please clarify for non-specialists that there is a wide range of e/ts combinations, reflecting non-steady bleaching profile, or something similar.*

Thank you for bringing this to our attention. The sentence has now been altered following your suggestions (line 368). We hope this is clearer to a non-specialist.

*Line 324, Figure 3. Please indicate the used exposure time for model without erosion in panels a and c.*

Good point, exposure time information has been added to both figures 3 and 4.

*Concerning panel d, since total exposure time used is historical data (so few tens of years), I don't understand how ts range can be explored between 0.1 and 10000 years with an output likelihood. If I understand well, ts <= exposure time, so there should be a large white (non-possible) area in panel d no?*

*Please justify the adopted approach, this is not really clear at present.*

We apologise for the lack of clarity in the original manuscript. The [10]Be data is the source of the white area as it represents the pairs of erosion and onset time which cannot reproduce the measured [10]Be concentrations. Panel 4(d) represents the results of one of the lower elevation samples where, as mentioned in the manuscript (lines 363-366), the [10]Be step is bypassed due to inheritance and archive information is used instead for exposure age information.

The upper $t_s$ value of the range is higher than the historical data because we used the same range as the other samples for consistency purposes. We believe it is sufficiently clear that the larger $t_s$ values are unrealistic for these samples (as the colour of those areas is associated with very low likelihood values).

In the updated manuscript, we have stated that the white areas are associated with values that are incompatible with the $^{10}$Be data (captions of Figures 3 and 4) and we hope that this makes it clearer.

*What is the red line on panels a and c (model with erosion)? The best-fitting parameter combination (maybe indicate with a star in b and d panels) or the region of high likelihood? Please clarify (same question for figure 4).*

The red line in these figures is created using values of $\dot{e}$ and $t_s$ that have a likelihood > 0.95. This information has been added to the captions of figures 3 and 4, as well as the main text (lines 382).

*Line 338-339. "When looking at the signals individually, the OSL125 and post-IR IRSL225 results reveal an anti-correlation between post-glacier erosion rates and elevation, whereas no trend is observed in the IRSL50 data (Fig. 5)". On Figure 5a one cannot differentiate the different signals (same symbols), can the authors change the symbols so that the reader can evaluate the differences?*

This is an excellent recommendation from the reviewer and we have now done this.

*Line 341. "Based on this, an average of the three signals was calculated for each site to generate one post-glacier erosion rate value".*

*I think this would be first interesting to discuss the different e/ts results between signals, before going to an average calculation. Is there some variability between signals in the output surface erosion rates? Why some signals appear in steady-state while other appear in transient state? I would think this is important for readers to have such information.*

*In addition, would it be possible to estimate some uncertainties (standard deviation? from likelihood?) and to show these on figure 5 for individual/averaged erosion rates?*

Thank you for the suggestion. We have added a section (4.1) that expands on the output erosion rate histories, highlighting the samples and signals which were not in steady state and hypothesising on reasons why. With regards to the uncertainties, we prefer not to include these in Figure 5 and readers can refer to Table 4 for uncertainties on the average erosion rates if they are interested in this information.

*Line 346. "minimum ts". There is no presentation of these outcomes in the section, I would suggest to provide more details about these and to confront them to total exposure time. For low-elevation samples, ts is close to exposure time, whereas it is really different (much lower) for high-elevation samples. I think this is important for the exposed results on lines 347-351, otherwise the readers could think longer exposure time = more eroded material...*

Thank you for the suggestion, however we prefer to leave this section as it is as to not overwhelm the readers. Anyone interested in more detail regarding the ts values can refer to Table 1, which highlights that although the values in lines (now) 386-390 are integrated, the higher elevations are experiencing lower erosion rates.

*Lines 368-370. It reads a bit strange to have the presentation of the slope relationship there (discussion), and not in the previous section along with the elevation relationship. Please consider presenting these in results too.*

We agree that it was strange to have these two interpretations in separate sections. The elevation relationship part has now been shifted to the discussions section, closer to the slope relationship information.

*Line 375. "local variations influencing the dominant post-glacier erosional mechanisms". Really vague, please specify what are those variations and mechanisms.*

*Alternatively, have the authors thought about potential correlation between erosion rate and exposure time? For Lehmann et al. (2020), the exposure times vary between ~20 ka and few years, while there the difference in exposure times is much lower. I agree that GG01 is not following this potential relationship, with a young exposure age, but given the different morphology/settings (cliff with periglacial erosion over 10s of ka), this may explain the low erosion rate.*

Thank you for your recommendation. Indeed, there are lithological and elevation differences between the two sites which are worth mentioning, and we have now expanded on this in the manuscript (lines 480-483) as well as the potential relationship between erosion rate and exposure time that is mentioned by the reviewer (lines 483-490).

*Lines 379-390. I agree that this is worth noting low bedrock surface erosion rates for such high-elevation environments, but these low erosion rates may also be the result of the sampling strategy, no? The sampling targets are specifically glacially-formed surfaces that are more or less preserved in the landscape, so they do reflect low surface erosion. I think that some further clarification could be given there.*

We do not understand the reviewer's point- these rates do not seem that low when compared to results from other studies (Figure 6). Furthermore, we do not feel the need to expand on the sampling strategy, as we believe it is clear enough earlier in the manuscript that these samples were taken from glacially-formed surfaces.

*Line 403. "...bedrock surface erosion rates from surfaces in glaciated environments, not currently subjected to glacial erosion,...". Reads a bit odd, please rephrase.*

Thank you for bringing this to our attention. We have rephrased the sentence (line 496).

*Lines 411-415. Are the referenced studies targeting bedrock/boulder surfaces that have been previously glaciated or not? Maybe this is important to specify. Same question for line 421 (" In Europe, Andrée (2022b)...").*

Some of the referenced studies targeted surface which been previously glaciated, while others not.

- Small et al. (1997): alpine bedrock summit surfaces that showed no evidence of past glaciations.
- Kirkbride and Bell (2010): glacial deposits.
- Nicholson (2008): from ice scoured bedrock surfaces.
- Sohbati et al. (2018): landslide and glacier erratic boulders.
- André (2002): roches moutonées and glaciofluvially scoured outcrops.
- Lehmann et al. (2019; 2020): previously glaciated bedrock surfaces.

As a result, we have altered each sentence to reflect this information.

*Line 415. "bedrock erosion rates". I thought Sohbati et al. (2018) only targeted boulders, please check.*

Indeed, they targeted boulders. Bedrock was written originally to reflect that they had collected rock samples, but following on from this comment, we have now changed the wording from "bedrock erosion rates" to "boulder erosion rates" (line 509) which is more accurate.

*Line 427. "these orders of magnitude are comparable with estimations of sub-glacial erosion rates and a summary of glacial and non-glacial erosion rates worldwide is displayed in Fig. 6". Have the authors tried to perform a pdf of the glacial and non-glacial erosion estimates. From visual inspection, I have the impression that glacial erosion rates, although they do overlap with non-glacial ones, are higher (and the presented scale is a log one!).*

*I appreciate this comparison and think that the compilation is interesting to discuss, however, I have a doubt about the actual comparison: "non-glacial rates" are apparently referring to "atmospheric" erosion/weathering and fluvial or landslide/hillslope erosion rates are not included right?*

*Then, what is really compared between these rates and glacial rates which do involve geomorphic agent as subglacial water/ice? I think this is important to clarify this point and justify why fluvial or landslide erosion rates (which are non-glacial agents) are not considered.*

Thank you for your useful comment. Unfortunately no, we have not performed a pdf of glacial and non-glacial estimates, and indeed the reviewer's observation that glacial erosion rates being higher than non-glacial erosion rates through visual inspection is correct. We have since added mean and standard deviation information to the figure to highlight this point and have also slightly altered the conclusions of the paper to reflect this.

With regards to our definition of "non-glacial", we have now changed this to "periglacial" (and thus removed all non-periglacial studies from the compilation figure). Fluvial and landslide erosion rates are still excluded because the studies mentioned are from glaciated environments.

*Line 444. "The large range is due to differences in sample locations...". How about differences in lithology (e.g. carbonate vs. crystalline bedrocks)?*

The role of lithology is an interesting point raised by the reviewer, however here all the references refer to studies on crystalline bedrock. Nevertheless, we have modified the sentence slightly to refer to one paper in particular where the differences in sample locations is drawn as a conclusion within the study itself.

*Line 462. "A full compilation of glacier erosion rates, calculations and methods can be found in Herman et al. (2021)". Maybe the authors can provide there the range in compiled glacial erosion rates?*

Thank you for the suggestion but we choose to keep the sentence as it is. We believe that Figure 6 is sufficient information, and the most relevant rates from the compilation are already mentioned in the main text. An interested reader can look to the referenced paper separately if they wish to know more.

*Line 484. "the dominant post-glacier erosion mechanisms". Please specify.*

The dominant post-glacier mechanism remains unclear; however we have expanded on the differences between the two sites (elevation and/or lithology) that could influence the erosional processes at play and included some words on the potential relationship that could exist between erosion rate and exposure time (as suggested by the reviewer earlier on in this review).

*I hope these comments and suggestions may be useful for revising the manuscript, and I look forward to seeing it published.*

*Sincerely,*

*Pierre Valla*

*Grenoble, 2 May 2022*

---

## Author Comment (AC3)

*Summary*

*This paper uses innovative $^{10}$Be-OSL measurements to derive erosion rates in the Swiss Alps during the Late Glacial and Holocene. It targets a vertical transect to assess the influence of elevation on erosion rates in this setting and shows that a negative correlation exists i.e. as the elevation decreases, the erosion rate increases. This is new and useful information as little is known about the factors that control erosion rates, especially in interglacial times. The erosion rates derived are similar in magnitude to existing studies, which gives confidence in the robustness of this new technique. Finally, the authors apply their data to address the long-standing uncertainties in our understanding of glacial vs non-glacial/interglacial erosion rates. Interestingly, their data suggests that interglacial erosion rates can be equally as important as glacial erosion rates in deglaciated environments, which is a key finding because this has important implications for understanding the drivers of rock erosion rates (e.g. climate) and thus, future rock erosion with anthropogenic climate change.*

*Overall, this is an excellent study, applying new techniques to a long-standing, challenging research question. The methods applied are robust, well justified and well performed. The study is generally well contextualised within the literature, but the understanding of the factors driving erosion rates could be better explained in the text in places (see specific comments) so it is easier for the reader to follow the authors interpretations. It was a very interesting read and I have some comments and questions below. It will be an excellent contribution to the literature in this area and I hope the comments are constructive.*

> We thank the reviewer for her positive feedback, helpful comments and appreciate her recognition of the relevance of our work. Her comments are addressed in detail in the sections below.

*General comments:*

1. *From my understanding, this is the first study to determine rock surface erosion rates using this technique using both K-feldspar and quartz, which is very important and interesting. The authors may wish emphasise this more in the intro/rationale/abstract, but I leave it to their discretion.*

> We thank the reviewer for her interest in our results, however since we don't have pure quartz/feldspar signals, we prefer not to expand on the matter as it would mostly be speculative.

2. *One of the advantages of the Lehmann et al. (2018) approach is that transient erosion rates can be derived, in addition to steady-state erosion rates. Given that this paper is focussed on interpreting the character and drivers of erosion, I would expect the authors to have more thoughts and interpretation of those samples that determine transient erosion, rather than just dismissing them as is stated in Line 324. For example, do these samples derive transient erosion rates because the technique/analysis is not reliable? Do these samples have different surficial characteristics than other samples? Is there any evidence of transient erosion for these samples (e.g. frost shattering) that is not present for the other samples? What natural*

*processes could have caused transient erosion in this setting? What even is transient erosion? This could be its own discussion point in the discussion before the steady-state erosion is discussed (Sections 4.1, 4.2).*

Thank you for the suggestion, we have added a section (4.1) that attempts to explain the steady state vs transient results. Unfortunately, we were unable to confidently identify the specific cause of all three signals being in a transient state with erosion for GG04. For the moment we have stated it is due to a "localised stochastic process" as any more detail would be purely speculative, although Fig. S1 suggests possible evidence of surface spallation.

However, for samples which only have either their $OSL_{125}$ or post-IR $IRSL_{225}$ signals in a transient state, we believe this could be due to the fact that they are more difficult to bleach, and as their bleaching profiles are thus necessarily closer to the surface, are therefore more susceptible to erosion and transient erosion states. This does not explain the transient $IRSL_{50}$ for sample GG03 which is interesting and requires further investigation that is beyond the scope of the present work.

3. *One of the main findings from this study is that "at present glacial erosion is assumed to have a greater influence on landscapes, yet a global compilation of both glacial and non-glacial erosion rates in deglaciated environments shows that erosion rates during interglacial times could be equally important" (Abstract, Lines 21-24). This is very interesting and is reflected in the data presented in this study. However, the discussion lacks discussion about glacial vs non-glacial or interglacial erosion rates. It could further unpack what natural processes differ between glacial and interglacial conditions that may or may not modulate the rock surface erosion (e.g. climate). Kirkbride and Bell (2010) do this well in the discussion of their study with respect to changing temperature and precipitation in glacial vs interglacial periods. Perhaps the discussion here could provide more insight into this as it is largely unknown due to the difficulty in determining glacial and interglacial erosion rates (i.e. deriving erosion rates on such resolution). The new data presented in this study on timeframes that were previously difficult to measure erosion rates on, therefore offers a great opportunity to explore these themes.*

Reading over the manuscript, we came to the realisation that our use of the word "glacial" was misleading and that it was unclear what exactly we were referring to. In fact, what we meant was subglacial erosion and we have now amended the wording to reflect this. Since we are looking at subglacial rates, which are influenced by the presence of ice, this makes it difficult to extrapolate to interpretations on the sub-aerial processes that differ between glacial and interglacial conditions. While it is a very interesting idea, to do so in this paper is beyond the scope of our research but we hope to see it included one day in a future study.

***Specific comments:***

*Please could the authors explain what they are referring to when they use the term "non-glacial erosion". Is it referring to the interglacial period (i.e. it has a time dimension) or a deglaciated setting (i.e. it has a space dimension)? It is a minor comment but it would help to*

*clarify this in the introduction before the reader continues on through the paper, perhaps around Line 34 where it is first mentioned.*

This is a good point. In this case, we are referring to any erosion in a glacial environment that is not related to glacial erosion (so a space dimension). Following on from this comment, we have added the sentence: "Here, non-glacial erosion refers broadly to any erosion occurring in a glacial environment that is not related to glacial erosion" (lines 38-39) for clarification purposes.

*Line 42 – here you refer to erosion studies during interglacial times and state that they are mainly limited to catchment-wide erosion rates but you could add 1-2 sentences to highlight that there are a few papers that have quantified interglacial erosion rates (e.g. Kirkbride and Bell, 2010; Sohbati et al. 2018; Lehmann et al. 2019; Smedley et al. 2021), which you will later expand upon in Section 1.1.*

Thank you for your recommendation, we have done this.

*Line 51-54 – it is useful to set up the aim of the study here, but I find it a little confusing that you report the main findings before presenting the data. Perhaps this is a feature of the journal and if so, that is fine as it is. If not, you might want to consider waiting to report the findings later in the paper.*

We understand why it might be confusing to report the main findings at this stage in the paper, however we prefer to keep the introduction as is.

*Line 81 - Smedley et al. (2021) also measured erosion rates over the last 4 ka so derived interglacial erosion rates and suggested that the transient nature of the erosion could have been caused by climate fluctuations over this time period. This is probably worth adding given the scarcity of papers that use TCN and OSL surface exposure methods to derive erosion rates.*

We thank the reviewer for bringing this paper to our attention and have now included a few sentences on this study to the paragraph (lines 89-92).

*Line 94 – technically Jenkins et al. (2018) performed burial dating, which is quite different from the exposure dating techniques mentioned. Discussing burial dating here is not necessary, but if you wish to demonstrate that it can be used for burial dating, I would be explicit about it and also add a reference to Freiesleben et al. 2015, for example: "In recent years, the application of OSL to rock surface dating has proved successful in a variety of settings for exposure dating (e.g. Sohbati et al., 2015; Liu et al., 2019; Lehmann et al., 2018) and burial dating (e.g. Freiesleben et al. 2015; Jenkins et al., 2018)."*

The reviewer raises a good point. We have now removed the Jenkins reference from this sentence.

*Line 100 – calibration for what? I suggest you add "after calibration to account for the rock-specific light attenuation rates" or something similar.*

Thank you for your detailed reading of the manuscript, we have amended the sentence so it reads "…after calibration to account for rock-specific bleaching rates" (line 116). We chose to use the term "bleaching" rather than "attenuation" as we were worried attenuation might be misleading to a reader who could think we were only calibrating for $\mu$ (the light attenuation parameter) which is not the case here.

*Line 105 – add reference to Smedley et al. (2021) as it is possibly the only other reference that has used multiple luminescence signals specifically for deriving rock erosion rates with 10Be and OSL measurements as you are doing in this study.*

Good point. We have now added this reference.

*Line 181 – please could you add a few words as to why you were sampling areas with minimal lichen cover and red, iron-oxide staining to explain to those who may wish to sample using this approach in the future. Why is it important?*

We have inserted the following ".. that would have otherwise impeded light penetration and impacted the luminescence signal" (line 199) at the end of the sentence.

*Line 188 – please could you add a line to explain why the approach of Elkadi et al. (2021) was beneficial for these measurements and so demonstrate the importance to the reader, e.g. does it dramatically improve the measurement reproducibility? Are the measurements more accurate?*

Thank you for your suggestion but we would rather not over-expand methodology. However, we have moved the Jenkins et al. (2018) reference to the more relevant part of sentence, so readers know which paper to refer to specifically if they want more information on the approach of Elkadi et al. (2021).

*Line 188 – it would be worth stating explicitly here that you will derive three signals per sample for comparison, so OSL signal of quartz, IR50 and pIRIR225 signals of feldspar. It would also be helpful to non-experts/users to explain why analysing multiple signals is useful in this context. It is really unique and interesting so worth emphasising.*

Excellent recommendation. We agree it would be beneficial for non-experts and so have added the relevant information in lines 207-210.

*Line 193 – subscript the n in Tn in both occurrences.*

Thank you to the reviewer for bringing this to our attention, we have now done this.

*Line 194 – please explain why the slices were excluded from further analysis? Does it mean the results would not be reliable? At present, to a non-expert the sentence makes it sound a little like they are just rejected and could be better explained (although very briefly!) why these criteria are applied.*

We have incorporated that monotonic signal decay is indicative of good heating (line 217) and also specified that slices which did not meet the criteria mentioned were excluded because they were not considered reproducible (line 218-219).

*Line 219 – here you may wish to also consider the work recently published by Furhmann et al. (2022) on the incidence angle of light given your interest in the orientation of the sample for calibration (https://doi.org/10.1016/j.radmeas.2022.106732).*

Thank you for bringing this study to our attention, we have added it to the paper.

*Line 222 – here you state that you have provided sample-specific calibration parameters by returning to each site after a year. Presumably this is for all three lithologies, so hornfels, schist and gneiss, AND for all three signals, which would be worth highlighting here for clarity.*

Thank you for highlighting that this might be unclear. We have now improved the sentence so it now reads: "…to calculate the unknown $\overline{\sigma\phi_0}$ and $\mu$ values for all three lithologies and luminescence signals" (lines 252-253) and we hope this makes it sufficiently clear.

*Given the infancy of the technique, the variability in lithology and the fact that you're using quartz and feldspar, I think this would be of great interest to the community and so would be worth including Table S1 into the main manuscript but this is the authors discretion.*

We understand why it could be beneficial to include the table in the manuscript but in the end, we feel it is better suited in the Supplementary. This is mostly due to the fact that, following on from the reviews, we have now expanded the information it contains and it has been split into 3 separate tables (Tables S2-S4) which we think is a lot to include in the main text.

*Lines 312-313 – it is unusual to include some interpretation in the results section but given that the discussion is focussed on the erosion rates rather than the specifics of the luminescence technique, it is reasonable. However, if you are going to offer some discussion of the OSL unknown parameters in Section 3.2, it would be useful to discuss how the quartz and feldspar attenuation rates compared given that no (or few) other examples exist in the literature showing such data and it would be interesting to unpack this unique data, especially relative to the variability in lithologies of the samples.*

We thank the reviewer for her interest in our results, but as mentioned in the general comments, we prefer not expand on this matter since we don't have pure quartz/feldspar signals although this is certainly worthy of further research.

*Line 356 – "Several factors, often working in combination with each other, modulate bedrock surface erosion rates. These include temperature, elevation and surface slope". This makes it sound like only three factors modulate erosion rates, which is not the case as explored by Portenga and Bierman (2011) amongst other studies. Presumably temperature, elevation and surface slope are factors you will focus on in this study? If so, either state all the factors that may modulate erosion and then say explicity that you'll only consider these three, or just re-phrase to "Several factors, often working in combination with each other, modulate bedrock*

*surface erosion rates. These include, but are not limited to, temperature, elevation and surface slope".*

Good point, we have rephrased the sentence as suggested.

*Line 364 – lithology is known to have a dominant control on rock surface erosion (e.g. Ford and Williams, 1989; Twidale, 1982; Moses et al. 2014), but this is not explicit from this section. It would be worth adding a sentence or two discussing the dominant role lithology has in modulating rock erosion rates, and then perhaps discussing whether you observe this in the erosion rates you measured for hornfels, schist and gneiss, or are they all similarly resistance to weathering and subsequent erosion? Given the metamorphic origin or these rocks, it is possible that they are more resistant than other lithologies (e.g. sandstones, limestones). Either way, it would be interesting having this discussion relative to your measured erosion rates, which are difficult to obtain.*

We agree that the effect of lithology was not sufficiently developed in the original manuscript and have now added a few sentences that we hope has done this (lines 422-428). However, we found no relationship between lithology and erosion rates for the samples in this study, likely due to the metamorphic nature of the samples as suggested by the reviewer. We have stated this in the manuscript (lines 428-431) while also adding a figure in the supplementary for additional information (Figure S5).

*Lines 379-390 – you state here that the anti-correlation between erosion rate and elevation is likely reflecting the lack of frost crack weathering in this setting, which is very interesting and new information, but where do your samples that derived transient erosion rates fit into this picture? Could these samples be reflecting frost crack weathering given that presumably frost cracking processes would be more stochastic over time and so more likely to be reflected by transient erosion, rather than steady-state. It would be interesting to have a better understanding of what transient erosion rates may be recording from the natural environment in general.*

As mentioned in the "General comments", while we agree with the reviewer that it would be very interesting to have a better understanding of what the transient erosion rates are representing with regards to the natural environment, we believe at this stage that the specific cause remains hypothetical and requires further investigation. We have added a section (4.1) that discusses the transient vs steady state erosion results, and in it we have included a sentence that says the transient state of sample GG04 is likely due to a localised stochastic process (line 400).

*Lines 391-394 – I find this a little confusing so perhaps you could better explain it for the reader. How do the observed patterns of glacial erosion in a valley due to quarrying and/or abrasion (that occur when the ice is present) control the interglacial erosion rates (when the ice is not present)? Are you suggesting that the rock has been weakened more during the glacial and so the interglacial erosion rates are higher at lower elevations? I think it would help the reader follow your arguments and interpretations better in this section if you provided a little more explanation for this.*

Yes, indeed this is what we were suggesting. Thank you for bringing to our attention that it might be unclear, we have now expanded the explanation and hope that it is clearer (lines 471-475).

*Line 399 – you give an example of frost crack weathering despite stating in Line 387-388 that "frost crack weathering is perhaps not a dominant form of post-glacier erosion in these areas", and rather "bedrock erosion is most likely occurring through continuous grain-by-grain erosion". I feel like these two interpretations do not align. Alternatively, have you considered the role of moisture via precipitation in this setting? Do lower elevations receive more rainfall/snowfall and therefore are subject to greater chemical weathering and subsequent erosion? It has long been known that precipitation can be a driver of rock weathering and subsequent erosion (e.g. Hall et al. 2012; Merill, 1906; Moses et al. 2014; Swantesson et al. 1992). Furthermore, in the 'global' compilation of rock outcrop erosion rates by Portenga and Bierman (2011), multi-variate statistical analysis showed that 32% of the variation in the global population of outcrop erosion rates could be explained by the five environmental parameters considered (latitude, elevation, relief, mean annual precipitation, mean annual temperature and seismicity), with mean annual precipitation being the most important parameter, accounting for 14% of the variability in this 'global' dataset even across many different settings. As such, it might be worth considering precipitation in your discussion. Although palaeo-precipitation records will be almost impossible, perhaps there are at least contemporary observational data of mean annual rainfall and snowfall from an elevation range of the alps for contextualisation?*

Thank you for bringing this contradiction in interpretation to our attention, and for raising this interesting precipitation hypothesis. With regards to the former point, we have removed the bracket that mentions frost crack weathering and for the latter, have now added precipitation as a potential explanation. While we weren't able to find contemporary observational data of mean annual rainfall and snowfall from an elevation range in the Alps, we investigated the Clausius-Clapeyron relationship which estimates a roughly 7% increase in water holding capacity of the atmosphere for every 1°C rise in temperature. The temperature difference between the lowest and highest elevation sites for this study is ~3.5°C, equating to a ~25% increase. We have included this information in the manuscript and expanded upon the calculation in the Supplementary Materials.

*Line 409 – Given the scarcity of studies, it is worth adding Smedley et al. 2021 as an OSL application, and then potentially expanding upon the findings of this study in Lines 411-425, given the authors determined interglacial erosion rates. Although the erosion rates derived were transient, it would be worth considering the erosion rates in the range that were lower and could be sustained for longer time intervals as these are more comparable to your steady-state erosion rates, in comparison to the higher erosion rates that can only be sustained over shorter timeframes.*

We agree that it is useful for a reader to have the Smedley et al. (2021) reference added to the sentence given the scarcity of OSL applications to erosion rates, and we have expanded upon its findings later in the manuscript (lines 519-524).